# Augment then Smooth:
# Reconciling Differential Privacy with Certified Robustness

**Jiapeng Wu**                                                                                    *paul@layer6.ai*
*Layer 6 AI*

**Atiyeh Ashari Ghomi**                                                                          *atiyeh@layer6.ai*
*Layer 6 AI*

**David Glukhov**                                                           *david.glukhov@mail.utoronto.ca*
*University of Toronto & Vector Institute*

**Jesse C. Cresswell**                                                                          *jesse@layer6.ai*
*Layer 6 AI*

**Franziska Boenisch**[*]                                                                     *boenisch@cispa.de*
*CISPA*

**Nicolas Papernot**                                                         *nicolas.papernot@utoronto.ca*
*University of Toronto & Vector Institute*

**Reviewed on OpenReview:** *https://openreview.net/forum?id=YNOIcnXqsr*

## Abstract

Machine learning models are susceptible to a variety of attacks that can erode trust, including attacks against the privacy of training data, and adversarial examples that jeopardize model accuracy. *Differential privacy* and *certified robustness* are effective frameworks for combating these two threats respectively, as they each provide future-proof guarantees. However, we show that standard differentially private model training is insufficient for providing strong certified robustness guarantees. Indeed, combining differential privacy and certified robustness in a single system is non-trivial, leading previous works to introduce complex training schemes that lack flexibility. In this work, we present DP-CERT, a simple and effective method that achieves both privacy and robustness guarantees simultaneously by integrating randomized smoothing into standard differentially private model training. Compared to the leading prior work, DP-CERT gives up to a 2.5× increase in certified accuracy for the same differential privacy guarantee on CIFAR10. Through in-depth per-sample metric analysis, we find that larger certifiable radii correlate with smaller local Lipschitz constants, and show that DP-CERT effectively reduces Lipschitz constants compared to other differentially private training methods. Code is available at github.com/layer6ai-labs/dp-cert.

## 1 Introduction

Machine learning (ML) models are becoming increasingly trusted in critical settings despite an incomplete understanding of their properties. This raises questions about the *trustworthiness* of those models, encompassing aspects such as privacy, robustness, and more. Society at large might expect *all* of these aspects to be accounted for simultaneously as ML's influence on everyday life expands, but scientists and practitioners still mostly grapple with each aspect individually.

---

[*]Work was done while the author was at the University of Toronto and the Vector Institute.

We aim to reconcile two key objectives of trustworthy ML, namely *privacy* and *robustness*. Privacy in the context of ML manifests as the requirement that a model does not leak information about the data it was trained on (Papernot et al., 2018), such as revealing whether or not certain data points were included in the training dataset (Shokri et al., 2017) or what characteristics they exhibit (Fredrikson et al., 2015). In our study, robustness refers to the requirement that a model's prediction should not change when its inputs are perturbed at test-time, even in the worst case of adversarially chosen perturbations.

The current gold standard for providing privacy guarantees is differential privacy (DP) (Dwork & Roth, 2014). In ML, DP produces mathematically rigorous privacy guarantees by limiting the impact of each training data point on the final model. This is achieved by clipping per-sample gradients, and adding a well-calibrated amount of noise to all model updates. Clipping limits the *sensitivity* of the training algorithm, while the addition of noise ensures that training will be more likely to output similar models when any individual data point is added to or removed from the training dataset.

To quantify robustness, we focus on test-time *certified robustness* (CR) (Wong & Kolter, 2018; Raghunathan et al., 2018), which provides probabilistic guarantees that perturbations of a certain magnitude will not change a model's prediction, regardless of what attack strategy is used to modify the inputs. A common approach for certifying robustness is *randomized smoothing*, where a classifier's outputs are averaged over a distribution surrounding the data point (Lecuyer et al., 2019; Li et al., 2019; Cohen et al., 2019).

Unfortunately, the two aims of providing DP and CR guarantees are in conflict, both empirically and conceptually. The clipping and noise addition used in DP training can impede the convergence of models (Tramèr & Boneh, 2021) and yield decision boundaries that are less smooth (Hayes et al., 2022), negatively impacting robustness (Fawzi et al., 2018). There is also substantial empirical evidence that providing privacy guarantees is in tension with robustness (Boenisch et al., 2021; Tursynbek et al., 2021). Integrating robustness measures into private training remains challenging because most methods to increase robustness use random or adversarial augmentations of training data points, which do not align well with DP training. Conceptually, augmenting an input increases the sensitivity of private training to it, and thereby provides additional avenues for information leakage. From a practical viewpoint, since gradients are computed on a per-example basis for DP, augmentations drastically increase the time and memory costs of training.

**Present Work.** We study the possible pitfalls of combining DP and CR. Through our analysis and ablation studies combining randomized smoothing techniques with DP training, we show that standard DP training of ML models is insufficient to provide strong CR results. Surprisingly, we find that prior complex methods used to regularize models to improve CR are unnecessary (Phan et al., 2019; 2020; Tang et al., 2022). Instead, we propose DP-CERT, a straightforward and adaptable framework for integrating CR into standard DP training which effectively incorporates augmentations while managing the additional privacy risks. Compared to the few existing approaches for integrating DP and CR, our proposed DP-CERT method is simpler, more flexible, and does not require extra network components. This decreased complexity is important for the widespread adoption of private and robust ML. DP-CERT even surpasses the state-of-the-art robustness guarantees on CIFAR10 under equivalent privacy guarantees.

To provide some insight into the mechanisms by which regularization methods improve CR for private models, we analyze CR on a per-data point basis. Using the gradient norm, Hessian spectral norm, and local Lipschitz constant, we find that the certifiable radius has a negative log-linear correlation with these quantities, and compare their distributions across methods. We conclude with concrete recommendations of best practices for the community to achieve CR and DP simultaneously.

## 2 Preliminaries

**Problem Setup.** We consider a $Y$-class classification task with a dataset $D = \{(x_i, y_i)\}_{i=1}^{n}$, where $x_i \in \mathbb{R}^d$ and $y_i \in \{1, ..., Y\}$. Let $f_\theta : \mathbb{R}^d \to \{1, ..., Y\}$ be a neural network classifier with parameters $\theta$, and $F_\theta$ denote the soft classifier which outputs the probability distribution, such that $f_\theta(x) = \arg\max_{y \in \{1,...,Y\}} F_\theta(x)_y$, where $F_\theta(x)_y$ denotes the model probability of $x$ being a member of class $y$.

**Differential Privacy and DPSGD.** We rely on the rigorous framework of differential privacy (DP) (Dwork et al., 2006) to obtain models with privacy guarantees. DP ensures that a model's weights at the end of training will be similar in distribution whether or not a particular data point was included in the training set. More formally, let $D$ and $D'$ be two potential training datasets for a model $f_\theta$ that differ in only one data point. The training mechanism $M$ guarantees $(\varepsilon, \delta)$-DP if, for all possible sets of outcomes $S$ of the training process, it holds that $\Pr\left[M(D) \in S\right] \leq e^\varepsilon \Pr\left[M(D') \in S\right] + \delta$. The parameter $\varepsilon$ specifies the privacy level, with smaller $\varepsilon$ yielding higher privacy, while $\delta$ quantifies the probability of a catastrophic failure of privacy (Kifer et al., 2022).

To obtain a differentially private variant of stochastic gradient descent (SGD), two modifications need to be made (Abadi et al., 2016). First, the individual gradients of each data point are clipped to a norm $C$ to limit the sensitivity of the model update caused by each data point. Second, choosing a noise level $\rho$, noise from $\mathcal{N}(0, \rho^2 C^2 \mathbf{I})$ is added to the aggregated gradients to prevent the changes to the model from revealing too much information about individual data points. The resulting algorithm DPSGD is detailed in Algorithm 1. We provide a more thorough introduction to DP and explanation of Algorithm 1 in Appendix A.1.

---

**Algorithm 1** Standard DPSGD, adapted from (Abadi et al., 2016).

---

**Require:** Private training set $D = \{(x_i, y_i) \mid i \in [N_{\mathrm{prv}}]\}$, loss function $L(\theta_t; x, y)$, Parameters: learning rate $\lambda_t$, noise scale $\rho$, group size $B$, gradient norm bound $C$.

1: **Initialize** $\theta_0$ randomly
2: **for** $t \in [T]$ **do**
3:     Sample mini-batch $B_t$ with sampling probability $B/N_{\mathrm{prv}}$ {Poisson sampling}
4:     For each $i \in B_t$, compute $\mathbf{g}_t(x_i) \leftarrow \nabla_\theta L(\theta_t; x_i, y_i)$ {Compute per-sample gradients}
5:     $\bar{\mathbf{g}}_t(x_i) \leftarrow \mathbf{g}_t(x_i)/\max\left(1, \frac{\|\mathbf{g}_t(x_i)\|_2}{C}\right)$ {Clip gradients}
6:     $\tilde{\mathbf{g}}_t \leftarrow \frac{1}{|B_t|}\left(\sum_i \bar{\mathbf{g}}_t(x_i) + \mathcal{N}\left(0, \rho^2 C^2 \mathbf{I}\right)\right)$ {Add noise to aggregated gradient}
7:     $\theta_{t+1} \leftarrow \theta_t - \lambda_t \tilde{\mathbf{g}}_t$ {Gradient descent}
8: **end for**
9: **Output** $\theta_T$ and compute the overall privacy cost $(\varepsilon, \delta)$ using a privacy accounting method.

---

**Certified Robustness.** Adversarial examples are a well-studied phenomenon in ML, in which an input to a model at test-time is perturbed in ways that do not alter its semantics yet cause the model to misclassify the perturbed input (Biggio et al., 2013; Szegedy et al., 2013; Goodfellow et al., 2014). Formally, for a given labeled datapoint $(x, y)$ and classifier $f$, an $(L_p, \zeta)$-adversary aims to create an adversarial example $x'$ such that $\|x' - x\|_p < \zeta$ and $f(x') \neq y$. Despite much research, the most common defense against adversarial examples remains adversarial training (Goodfellow et al., 2014; Zhang et al., 2019). While adversarial training improves robustness to known algorithms for finding adversarial examples, it does not guarantee that a model will be robust to all adversarial examples. This motivates developing certifiable guarantees of robustness which provide a lower bound $r$ on the distance between a correctly classified input and any adversarial example that may be misclassified (Wong & Kolter, 2018; Raghunathan et al., 2018). This lower bound is also known as the certification radius.

**Randomized Smoothing.** A popular approach for establishing certified robustness (CR) guarantees is through probabilistic robustness verification which, with high probability, verifies that no adversarial examples exist within a radius $r$ of the original input (Li et al., 2023). The most common method smooths a classifier by averaging the class predictions of $f$ using a smoothing distribution $\mu$ (Lecuyer et al., 2019; Li et al., 2019; Cohen et al., 2019),

$$\hat{g}(x) = \arg\max_{c \in [Y]} \int_{\zeta \in \mathrm{supp}(\mu)} \mathbb{I}[f(x + \zeta), c]\mu(\zeta)\, d\zeta, \tag{1}$$

where $\mathbb{I}[a, b] = 1 \iff a = b$ and 0 otherwise (Li et al., 2023). As computing the integral in Equation (1) is intractable, Monte Carlo sampling is used. We denote the approximation of $\hat{g}$ given by Monte Carlo sampling as $g$. Smoothed classifiers are evaluated in terms of their certified accuracy—the fraction of samples correctly classified when certifying robustness at a given radius $r$.

A tight $L_2$ radius was obtained by Cohen et al. (2019) when using isotropic Gaussian noise $\mu = \mathcal{N}(x, \sigma^2 \mathbf{I})$, where $\sigma$ is a hyperparameter that controls a robustness/accuracy tradeoff. In particular, it was proved that for any base classifier $f$, the Gaussian smoothed classifier $g$ is robust around an input $x$ with radius $r = \frac{\sigma}{2}(\Phi^{-1}(p_A) - \Phi^{-1}(p_B))$ where $p_A$ and $p_B$ denote the probabilities of $c_A$ and $c_B$, the most and second-most probable classes returned by $g(x)$, and $\Phi^{-1}$ is the inverse of the standard Gaussian CDF. The exact probabilities $p_A$ and $p_B$ are not needed; one can use lower $\underline{p_A} \leq p_A$ and upper $\overline{p_B} \geq p_B$ bounds instead, approximated by Monte Carlo sampling. The output of the smoothed classifier $g(x)$ is approximated by aggregating the predictions of a base classifier $f(x + \eta)$ for $\eta \sim \mathcal{N}(0, \sigma^2 \mathbf{I})$. As a high dimensional standard Gaussian assigns almost no mass near its mean, ensuring that $g(x)$ is accurate at large certification radii $r$ requires $f$ to be accurate on Gaussian perturbed data (Gao et al., 2022).

**Related Work.** Previous works combining DP and CR deviate significantly from the standard DPSGD template, and as a result become more difficult to integrate into DP training pipelines.

Phan et al. (2019) introduced Secure-SGD, the first framework aimed at achieving both CR and DP simultaneously. They used a PixelDP approach (Lecuyer et al., 2019) to attain CR and introduced the heterogeneous Gaussian mechanism, which adds heterogeneous noise across elements of the gradient vector.

Phan et al. (2020) introduced StoBatch which employs an autoencoder (Hinton & Zemel, 1993) and a functional DP mechanism, objective perturbation (Zhang et al., 2012), to reconstruct input examples while preserving DP. Subsequently, reconstructed data is used to train a deep neural network, along with adversarial training (Tramèr et al., 2017) to improve robustness.

Tang et al. (2022) proposed perturbing input gradients during training and introduced the multivariate Gaussian mechanism which allows them to achieve the same DP guarantee with less noise added. Their method TransDenoiser follows the architecture of denoised smoothing, adding differentially private noise to a pre-trained classifier.

We note that these prior approaches either add trainable model components increasing the overall complexity, or lack the flexibility to incorporate the latest adversarial training methods.

## 3 Method

Training ML models to be both differentially private and certifiably robust poses two main challenges. In this section, we describe these challenges and how our DP-CERT framework meets them by effectively improving certified robustness while preserving privacy.

The first challenge surfaces around the use of adversarial training or augmentations with DPSGD. As shown by Cohen et al. (2019), data augmentations used for training can enhance a model's CR, however, augmented data points could leak private information about the originals. Previous works combining DP and CR have proposed adding noise or adversarial examples during training, but deviate from the standard DPSGD template to address the privacy risks (Phan et al., 2019; 2020; Tang et al., 2022).

The second challenge is that gradient clipping and noise addition in DPSGD harms the convergence rate of training (Chen et al., 2020; Tramèr & Boneh, 2021; Bu et al., 2023a), while restrictive privacy budgets may require stopping training prior to convergence. Robustness on the other hand suffers when models are not converged, as having large gradients at test points makes finding adversarial examples easier (Fawzi et al., 2018). Strategies must be employed to improve the convergence of private training, or regularize gradient size.

We aim to make CR feasible within the standard pipeline of DPSGD, with state-of-the-art convergence and proper accounting for additional privacy risks. Our DP-CERT framework effectively trains with augmented samples while preserving privacy, and integrates advancements in adversarial training and regularizers to enhance certifiable robustness (Salman et al., 2019; Li et al., 2019; Zhai et al., 2020). A schematic of DP-CERT is given in Figure 1 and the training framework in described in Algorithm 2 below. We describe the components in turn.

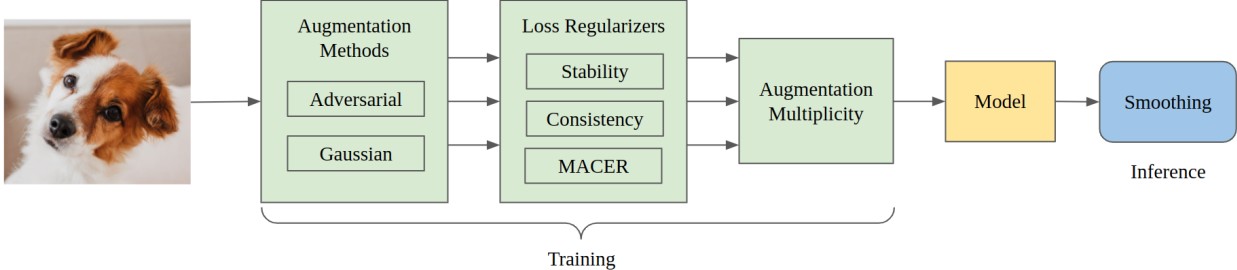

Figure 1: The DP-CERT training framework for providing strong CR guarantees within DPSGD.

**Augmentation Multiplicity.** The first step in Figure 1 involves training on augmented data with the aim of improving robustness. For each data point $(x_i, y_i)$, we obtain $K$ augmented data points $(x_i^j, y_i)$, where $j \in \{1, ..., K\}$ and $x_i^j$ is the $j$-th augmented version of $x_i$. For notational convenience, we use $x_i^0$ to denote the original data point $x_i$. To improve CR, augmentations may be generated adversarially, but since this is computationally expensive, we also consider Gaussian perturbations which, as shown by Cohen et al. (2019), also can enhance a model's certified robustness. When using Gaussian perturbations instead of adversarial training, we define $x_i^j = x_i + \eta_j$, $\eta_j \sim \mathcal{N}(0, \sigma^2 \mathbf{I})$ for $j \neq 0$.

With respect to the first challenge, an important component of DP-CERT is how we handle the privacy concerns of training with augmented data points. We employ augmentation multiplicity, introduced by De et al. (2022) as a method for incorporating standard augmentation methods into DPSGD training. The method involves averaging the gradients of multiple input augmentations of the same training sample before clipping and noising. Since all downstream impact to the model weights from sample $x_i$ is contained in this averaged gradient, clipping it provides a finite sensitivity as required for the sampled Gaussian mechanism used in DPSGD (Mironov et al., 2019), and no additional privacy cost is incurred. We propose to apply a slightly modified version of this method for improving CR with DPSGD, namely, by leveraging augmentation multiplicity to train the network with Gaussian noised input data, akin to classical methods of regularizing models for CR (Cohen et al., 2019).

The model updates can be expressed as follows

$$\theta_{t+1} = \theta_t - \lambda_t \left[ \frac{1}{|B_t|} \sum_{i \in B_t} \text{clip}_C \left( \frac{1}{K+1} \sum_{j=0}^{K} \nabla_\theta L(\theta_t; x_i^j, y_i) \right) + \frac{\rho C}{|B_t|} \xi \right]. \tag{2}$$

$\theta_t$ denotes the model parameters at iteration $t$, $\lambda_t$ is the learning rate, $B_t$ is the $t$'th batch, $C$ is the clipping bound, $K$ is the number of augmentations, $\rho$ is the noise multiplier, $\xi \sim \mathcal{N}(0, \mathbf{I})$, and $\nabla_\theta L(\theta_t; x_i^j, y_i)$ is the gradient of the loss for data point $(x_i^j, y_i)$. Note that $j$ starts from 0, which means we include the *original samples* along with the augmented ones in model training. We discovered that including the original samples empirically improves the robust accuracy.

**Adversarial Training.** When not using Gaussian augmentations to achieve better certified accuracy, we incorporate adversarial training by deploying existing attacks to create adversarial examples. Specifically, we integrate SmoothAdv (Salman et al., 2019) into private training, which, given original data $(x, y)$, optimizes

$$\underset{\|x'-x\|_2 \leq \epsilon}{\arg\max} \left( -\log \underset{\eta \sim \mathcal{N}(0, \sigma^2 I)}{\mathbb{E}} [F_\theta(x' + \eta)_y] \right), \tag{3}$$

to find an $x'$ $\epsilon$-close to $x$ that maximizes the cross entropy between the smoothed classifier $g_\theta(x')$, and label $y$. Using Monte Carlo sampling, Objective (3) can be optimized by iteratively computing the approximate gradient

$$\nabla_{x'} \left( -\log \left( \frac{1}{K} \sum_{j=1}^{K} F_\theta(x' + \eta_j)_y \right) \right). \tag{4}$$

where $\eta_1, ..., \eta_K \sim \mathcal{N}(0, \sigma^2 \mathbf{I})$. The approximate gradient is then used to update $x'$, with the final $x'$ used as examples within augmentation multiplicity.

**Regularization.** To address the second challenge, the second step in Figure 1 adapts stability regularization to private training in order to minimize the distance between the output probability of the original and augmented examples, thereby improving the robustness to input noise. Stability training (Li et al., 2019) adds a smoothed cross-entropy loss as regularization. Inspired by TRADES (Zhang et al., 2019), we instead use the Kullback–Leibler (KL) divergence with a hyperparameter $\gamma$ controlling the strength of the regularization as:

$$L_{\text{stab}}(\theta; x_i, y_i) = \sum_j L(\theta; x_i^j, y_i) + \gamma D_{\text{KL}}\big(F_\theta(x_i) \,\|\, F_\theta(x_i^j)\big). \tag{5}$$

Additionally, we propose integrating MACER (Zhai et al., 2020), an alternative training modification to directly optimize the certified accuracy at larger robustness radii without requiring the costly process of adversarial training. MACER achieves this by decomposing the error of a smoothed classifier into a classification error term and a robustness error term, the latter reflecting whether or not the smoothed classifier is able to certify robustness for a given radius. MACER is further described in Appendix B.1.3.

---

**Algorithm 2** DP-CERT Training

---

**Require:** Private training set $D = \{(x_i, y_i) \mid i \in [N_{\text{prv}}]\}$, loss function $L(\theta_t; x, y)$, Parameters: learning rate $\lambda_t$, noise scale $\rho$, group size $B$, gradient norm bound $C$, number of augmentations $K$.

1: **Initialize** $\theta_0$ randomly
2: **for** $t \in [T]$ **do**
3:      Sample mini-batch $B_t$ with sampling probability $B/N_{\text{prv}}$ {Poisson sampling}
4:      For each $i \in B_t$, generate $K$ augmented samples $x_i^j$ {Adversarial or Gaussian augmentations}
5:      $L'(\theta_t; x_i, y_i) \leftarrow \frac{1}{K+1} \sum_{j=0}^{K} L(\theta_t; x_i^j, y_i) + R(\theta_t, x_i^j)$ {Add regularizer, average over augmentations}
6:      $\mathbf{g}_t(x_i) \leftarrow \nabla_\theta L'(\theta_t; x_i, y_i)$ {Per-sample gradients contain all information from $x_i$ and its augmentations}
7:      $\bar{\mathbf{g}}_t(x_i) \leftarrow \mathbf{g}_t(x_i)/ \max\left(1, \frac{\|\mathbf{g}_t(x_i)\|_2}{C}\right)$ {Clip gradients}
8:      $\tilde{\mathbf{g}}_t \leftarrow \frac{1}{|B_t|}\left(\sum_i \bar{\mathbf{g}}_t(x_i) + \mathcal{N}\left(0, \rho^2 C^2 \mathbf{I}\right)\right)$ {Add noise to aggregated gradient}
9:      $\theta_{t+1} \leftarrow \theta_t - \lambda_t \tilde{\mathbf{g}}_t$ {Gradient descent}
10: **end for**
11: **Output** $\theta_T$ and compute the overall privacy cost $(\varepsilon, \delta)$ using a privacy accounting method.

---

**Summary of Method.** DP-CERT (Figure 1, Algorithm 2) combines augmentation multiplicity, which was originally introduced to improve the *accuracy* of models trained by DPSGD, with randomized smoothing for inference, and a variety of adversarial augmentation and regularization approaches, which have been shown to improve certified robustness. Comparing the training algorithms 1 and 2 shows that our method follows the template of

Table 1: Instantiations of DP-CERT

| Method | Augmentation | Regularization |
|---|---|---|
| DP-Gaussian | Gaussian | None |
| DP-SmoothAdv | Adversarial | None |
| DP-Stability | Gaussian | Stability |
| DP-MACER | Gaussian | MACER |

DPSGD with the following modifications: adversarial or Gaussian perturbations are generated for each data point in the batch; regularization is optionally applied to the loss function; and augmentation multiplicity is used to compute "per-sample" gradients as the average over all augmented data points associated to $x_i$. Randomized smoothing at inference time allows certifying robustness, but improved robustness is a result of the training-time modifications.

While DP-CERT is a combination of known methods we emphasize its simplicity compared to previous works that bring CR guarantees to DP. Decreasing complexity is vital for the widespread adoption of robust and private AI, and we simultaneously report improved performance.

**Privacy Guarantee.** As described above, augmentation multiplicity achieves the same privacy guarantees as DPSGD by averaging over all augmentations before clipping gradients. This remains true whether random or adversarial augmentations are used, and for any regularized loss function. Once a model is trained with a DP guarantee, it can be used for inference with randomized smoothing at no further privacy cost, thanks to the postprocessing guarantee of DP (Dwork & Roth, 2014; Abadi et al., 2016). Hence, DP-CERT requires no special accounting, and is easily implemented within any mature code package for DP. We note that Gaussian augmentations are a form of input perturbation and may contribute to stronger privacy guarantees. However, input perturbation would first require data points to be clipped for finite sensitivity, impacting performance. Also, we keep the original datapoint that does not have noise added, which is beneficial to performance but not aligned with input perturbation. Hence, we do not attempt to account for DP guarantees from this source of randomness.

In the following sections, we experimentally benchmark four instantiations of the DP-CERT framework as summarized in Table 1.

## 4 Experiment Setup

We evaluate the effectiveness of DP-CERT on multiple image classification datasets, including MNIST (Le-Cun et al., 2010), Fashion-MNIST (Xiao et al., 2017), and CIFAR10 (Krizhevsky & Hinton, 2009). More detailed data statistics can be found in Appendix C.1. We demonstrate that DP-CERT consistently outperforms the undefended differentially private baselines, and establishes the state of the art for certified $L_2$ defense under a DP guarantee on CIFAR10.

### 4.1 Baseline Methods

Our comparisons include non-private training (*Regular*) for an accuracy baseline, along with *DPSGD*, and per-sample adaptive clipping (*PSAC*) (Xia et al., 2023) which is a variation of DPSGD with better convergence (see Appendix A.2 for details). For DPSGD and PSAC, we adopt the same settings as Xia et al. (2023), who exhibited state-of-the-art performance for DP optimization on various tasks. These baselines show what CR guarantees can be achieved by DP models when combined with randomized smoothing, but without training-time modifications designed to protect against adversarial examples.

We additionally compare against prior approaches to integrate CR with DP guarantees, namely SecureSGD (Phan et al., 2019), StoBatch (Phan et al., 2020), and TransDenoiser (Tang et al., 2022) on CIFAR10. However, the lack of open source implementations in Pytorch combined with the complexity of these approaches prevents us from running experiments with them. Hence, we directly quote the results published in (Tang et al., 2022) for the three baselines, and replicate their experimental setting to run our methods. We again note that our approach is much simpler, fitting the standard DPSGD training pipeline that is supported by many mature code packages, and we provide our code at github.com/layer6ai-labs/dp-cert.

For all methods that guarantee privacy (i.e. all methods other than Regular), we use models that achieve the same level of privacy ($\varepsilon = 3.0, \delta = 10^{-5}$) to ensure a fair performance comparison.

### 4.2 Implementation and Hyperparameters

For all experiments on MNIST and Fashion-MNIST, we train a four-layer CNN model from scratch, with the settings used by Tramèr & Boneh (2021). On CIFAR10, we fine-tune a CrossViT-Tiny (Chen et al., 2021), pretrained on ImageNet1k (Deng et al., 2009).

For each model configuration, we consider three models trained with different noise levels $\sigma \in \{0.25, 0.5, 1.0\}$ for smoothing at training time, and during inference we apply randomized smoothing with the same $\sigma$ as used in training.

TransDenoiser fine-tunes a pretrained VGG16 model (Simonyan & Zisserman, 2014) and uses an additional denoising diffusion model. For a fair comparison, we fine-tune a much smaller network for DP-CERT, a

CrossViT-Tiny (Chen et al., 2021). For additional implementation and hyperparameter settings, we refer the reader to Appendix C.3.

### 4.3 Evaluation Metrics

First, we report the clean accuracy (*Acc*) on the test dataset without randomized smoothing for inference as a measure of convergence. Following previous works, we report the *approximate certified accuracy*, which is the fraction of the test set that can be certified to be robust at radius $r$ using the CERTIFY procedure introduced by Cohen et al. (2019). We also include the average certified radius (*ACR*) (Zhai et al., 2020) returned by CERTIFY which serves as an additional metric for better comparison of CR between two models (Tsipras et al., 2019; Zhang et al., 2019). ACR is calculated as

$$\text{ACR} = \frac{1}{|D_\text{test}|} \sum_{(x,y) \in D_\text{test}} \text{CR}(f, \sigma, x) \cdot \mathbb{I}[\hat{f}(x), y'], \tag{6}$$

where $D_\text{test}$ is the test dataset, and CR denotes the certified radius provided by CERTIFY.

## 5 Experimental Evaluation

Table 2: Comparison of accuracy, ACR and the certified accuracy at radius $r = 0.25$ between baselines and instances of the DP-CERT framework on MNIST, Fashion-MNIST and CIFAR10. Higher is better for all metrics.

| $\sigma$ | Method | MNIST | | | Fashion-MNIST | | | CIFAR10 | | |
|---|---|---|---|---|---|---|---|---|---|---|
| | | Acc | ACR | $r{=}0.25$ | Acc | ACR | $r{=}0.25$ | Acc | ACR | $r{=}0.25$ |
| 0.25 | Regular | 99.14 | 0.581 | 83.6 | 89.28 | 0.359 | 55.5 | 94.79 | 0.055 | 9.5 |
| | DPSGD | 98.13 | 0.606 | 88.3 | 85.87 | 0.343 | 53.2 | 89.74 | 0.023 | 3.3 |
| | PSAC | **98.25** | 0.608 | 88.5 | **86.34** | 0.320 | 49.0 | **89.81** | 0.020 | 2.8 |
| | DP-Gaussian | 98.13 | 0.735 | 95.7 | 84.76 | 0.545 | 75.8 | 87.61 | 0.246 | 41.8 |
| | DP-SmoothAdv | 98.08 | **0.742** | **96.0** | 83.97 | **0.554** | **75.9** | 87.89 | **0.275** | **44.3** |
| | DP-Stability | 97.86 | 0.738 | 95.9 | 84.19 | 0.551 | 75.7 | 88.53 | 0.246 | 41.6 |
| | DP-MACER | 98.13 | 0.736 | 95.6 | 84.79 | 0.545 | 75.8 | 87.52 | 0.246 | 41.7 |
| 0.5 | Regular | 99.14 | 0.308 | 31.8 | 89.28 | 0.331 | 34.9 | 94.79 | 0.092 | 9.7 |
| | DPSGD | 98.13 | 0.344 | 50.0 | 85.87 | 0.309 | 29.8 | 89.74 | 0.057 | 9.8 |
| | PSAC | **98.25** | 0.383 | 55.9 | **86.34** | 0.298 | 27.5 | **89.81** | 0.056 | 9.8 |
| | DP-Gaussian | 97.74 | 1.246 | 94.7 | 82.42 | 0.879 | 73.0 | 87.48 | **0.288** | **35.5** |
| | DP-SmoothAdv | 97.66 | **1.258** | **94.8** | 82.65 | **0.894** | 73.0 | 87.54 | 0.263 | 31.9 |
| | DP-Stability | 97.62 | 1.248 | 94.6 | 82.25 | 0.876 | 72.7 | 88.56 | 0.282 | 35.4 |
| | DP-MACER | 97.75 | 1.246 | 94.7 | 82.50 | 0.880 | **73.1** | 87.36 | 0.287 | 35.2 |
| 1.0 | Regular | 99.14 | 0.257 | 10.7 | 89.28 | 0.342 | 21.2 | 94.79 | 0.079 | 9.7 |
| | DPSGD | 98.13 | 0.260 | 10.4 | 85.87 | 0.338 | 13.5 | 89.74 | 0.029 | 5.9 |
| | PSAC | **98.25** | 0.213 | 20.0 | **86.34** | 0.328 | 11.5 | **89.81** | 0.023 | 4.2 |
| | DP-Gaussian | 96.33 | **1.262** | **85.6** | 80.96 | 1.101 | 65.4 | 88.55 | **0.299** | **25.4** |
| | DP-SmoothAdv | 96.54 | 1.249 | 85.0 | 80.93 | 1.096 | 64.7 | 87.37 | 0.237 | 21.0 |
| | DP-Stability | 96.48 | **1.262** | 84.9 | 80.66 | 1.084 | 65.1 | 89.09 | 0.294 | 25.0 |
| | DP-MACER | 96.31 | **1.262** | 85.5 | 80.83 | **1.102** | 65.3 | 88.40 | 0.255 | 22.3 |

In Table 2, we compare our baseline methods and DP-CERT instantiations by their clean accuracy, ACR, and certified accuracy for radius $r = 0.25$. The best ACR and certified accuracy for each smoothing radius $\sigma$ are displayed in **bold**, while close runner-ups are underlined. In Figure 2 we plot the certified accuracy as the certification radius is increased on CIFAR10 (See Figure 7 in Appendix D.1 for MNIST and Fashion-MNIST). We also compare DP-CERT to SecureSGD, StoBatch, and TransDenoiser on CIFAR10 in Figure 3, copying the baseline results from (Tang et al., 2022).

**Discussion.** Table 2 and Figure 2 show that all instantiations of DP-CERT significantly outperform the baseline methods in terms of the approximate certified accuracy and ACR. Generally, DP-CERT's clean accuracy is marginally lower than the PSAC baseline, but its ACR and certified accuracy do not fall off drastically as $\sigma$ is increased. Hence, for private models there is still a tradeoff between clean accuracy and

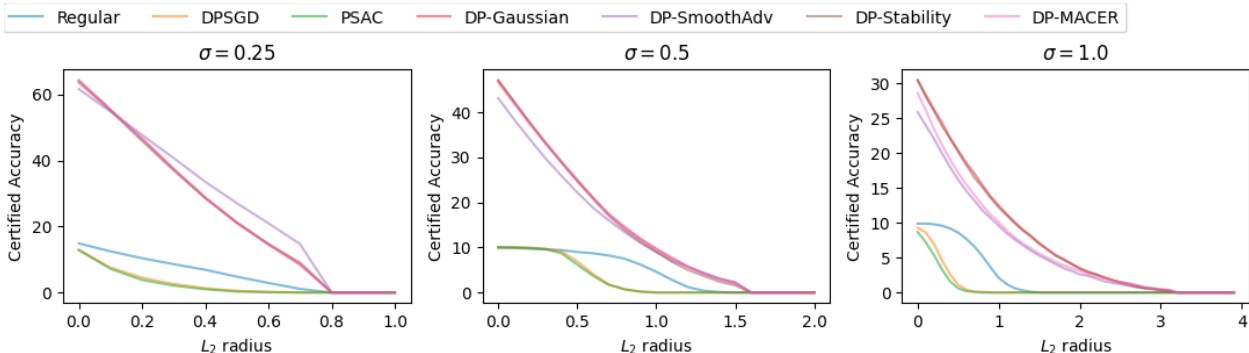

Figure 2: Approximate certified accuracy (ACR) comparison on CIFAR10.

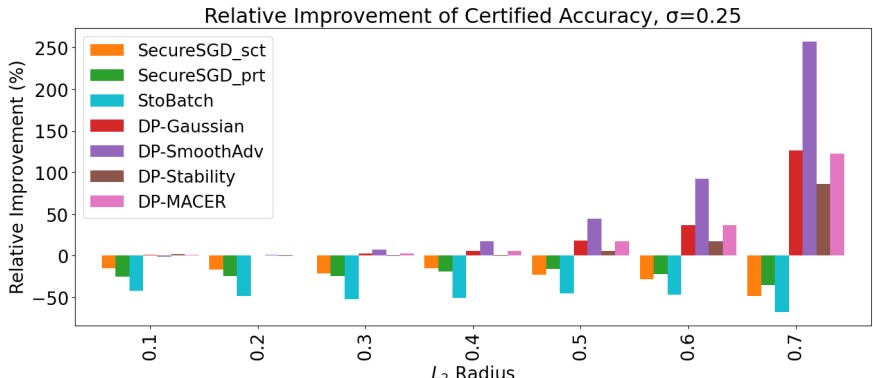

Figure 3: Approximate certified accuracy comparison on CIFAR10. The y-axis shows the relative certified accuracy improvement over the strongest baseline method, TransDenoiser, as percentages. Baseline results are from (Tang et al., 2022).

CR, as there is for non-private models. Our well-converged baselines show that DPSGD training does not always lead to worse CR compared to non-private training, which should be contrasted with previous studies on the adversarial robustness of DPSGD (Tursynbek et al., 2021; Boenisch et al., 2021; Zhang & Bu, 2022). Still, DPSGD alone, even when well-converged, does not provide strong CR via randomized smoothing, demonstrating the need for DP-CERT's improvements.

Figure 3 shows that all variants of DP-CERT surpass the state-of-the-art certified accuracy on CIFAR10 with a much smaller pre-trained model compared to (Phan et al., 2019), (Phan et al., 2020), and (Tang et al., 2022). At higher certification radii, for instance an $L_2$ radius of 0.7, the best version of DP-CERT is over 250% better than TransDenoiser. Among baselines, TransDenoiser is less than 40% better than the next strongest method, Secure-SGD, showing that DP-CERT is a meaningful advancement. Since we do not rely on an additional denoiser, inference is also much faster with DP-CERT.

**Practical recommendations from Table 2.** Contrary to prior findings in non-private training (Salman et al., 2019; Zhai et al., 2020), all methods we tested to improve certified robustness have similar performance when used with differentially private training as instantiations of DP-CERT (Table 1). Because DP-SmoothAdv incurs a significant training overhead from adversarial perturbations, we recommend *not* using it in private training. Gaussian augmentations are less expensive, and give just as good performance. For training from scratch, *DP-Gaussian* is recommended since it offers competitive results while being the most straightforward to implement and fastest to train. For fine-tuning a pre-trained model, *DP-Stability* is recommended since it has the highest clean accuracy in all variants while offering competitive certified accuracy.

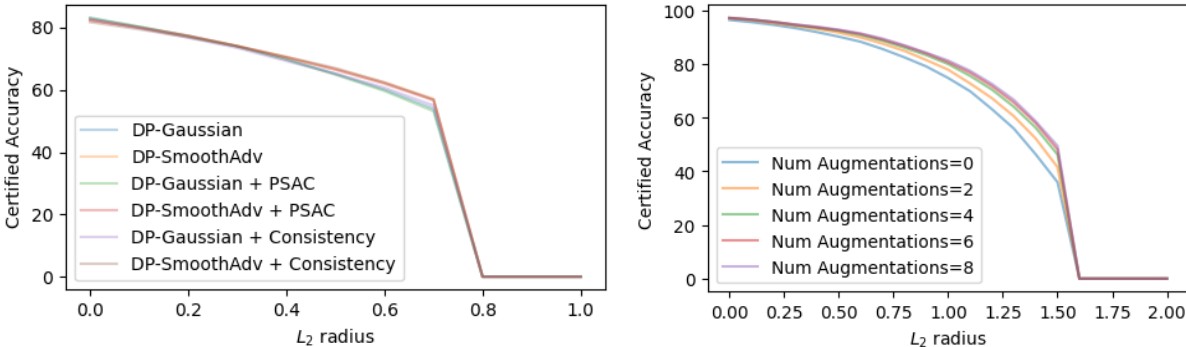

Figure 4: Ablation for consistency regularization, PSAC, and augmentation number on Fashion-MNIST.

### 5.1 Impact of Model Architectures and Augmentations

In this section, as an ablation, we examine the effect of different model variants and hyperparameters. All experiments are run on Fashion-MNIST with $\sigma = 0.5$; results for MNIST and other values of $\sigma$ are given in Appendix D.2 and confirm the trends shown here. We combine consistency regularization (reviewed in Appendix B.1.2) and PSAC with DP-Gaussian and DP-SmoothAdv to study their effect on certified accuracy and radius. Figure 4 (left) shows that neither of these techniques improves CR. We emphasize that the simplest training method, DP-Gaussian, performs just as well without specialized techniques designed to enhance convergence and robustness in other contexts. We also train models with different numbers of augmentations and compare their CR. Figure 4 (right) shows that the certified test accuracy is largely unchanged as the number of augmentations increases, consistent with the observations made by Salman et al. (2019). Note that for this plot, when the number of augmentations is zero we replace the original sample with a version corrupted by Gaussian noise to benefit CR (the version without augmentation of any kind is simply DPSGD which has much worse CR, as shown in Figure 7). Since using fewer augmentations better preserves the natural accuracy and incurs less training overhead, we recommend using a minimal number of augmentations.

We provide additional hyperparameter ablation results across learning rates $\lambda$ and clipping norms $C$ in Appendix D.2. The results show that our settings achieve the highest clean accuracy, and that accuracy is fairly stable around these optimal parameters.

### 5.2 Per-sample Metric Analysis

Certified accuracy and ACR are both metrics averaged over the test set. However, robustness is inherently sample based, since it examines how resilient the model is against perturbations tailored to individual samples. Therefore, in this section we use per-sample metrics of robustness and conduct an in-depth analysis of their distributions to provide insights into why certain training methods may produce more robust models than others.

We consider three per-sample metrics associated with robustness: the *input gradient norm*, *input Hessian spectral norm*, and *local-Lipschitz constant*. The first two metrics measure the local smoothness of the loss landscape with respect to the input space. Taylor's approximation can be used to show a direct link between these two metrics and the worst-case change in loss from small input perturbations. Due to this connection, prior works directly regularized them to improve robustness (Hoffman et al., 2019; Jakubovitz & Giryes, 2018; Moosavi-Dezfooli et al., 2019).

Gradients and Hessians are highly local quantities that are only connected to robustness through Taylor's approximation at small radii around the input data point. Consequently, they may not be informative at larger radii used to certify robustness. Thus, we also compare models using an empirical estimate of the average local Lipschitz constant of the model's penultimate layer. By viewing the network as a feature extractor composed with a linear classifier, using the penultimate layer captures the worst-case sensitivity of

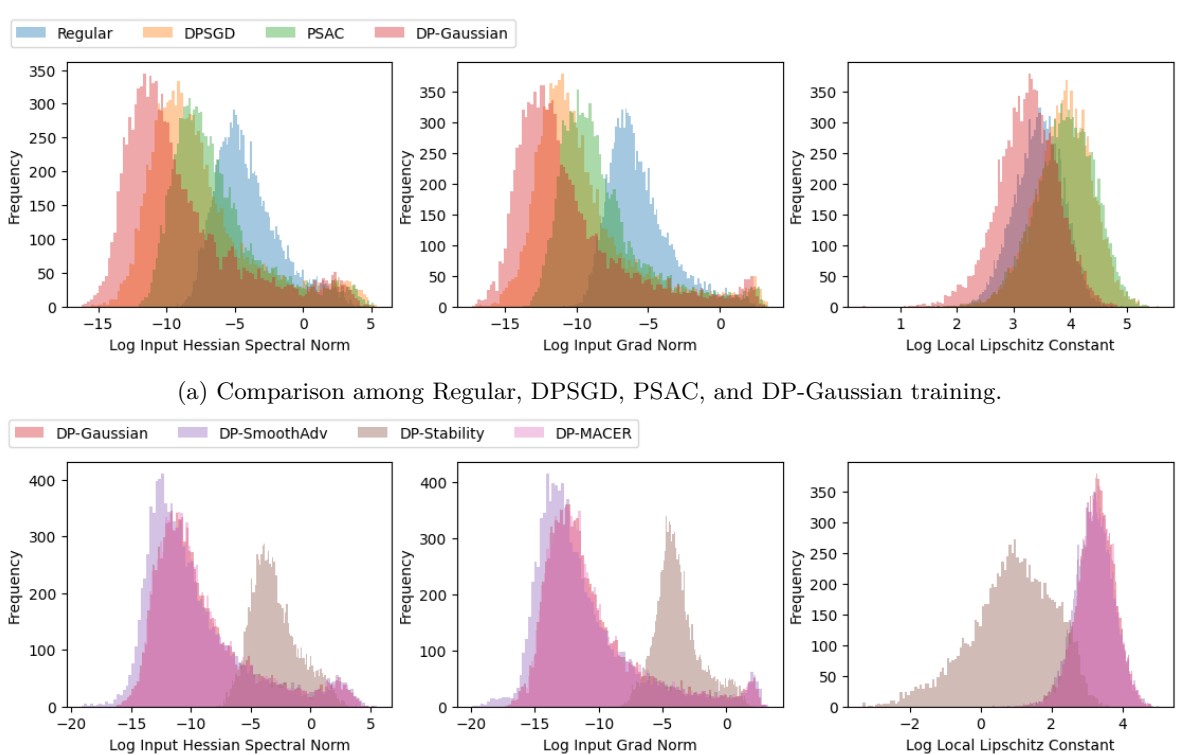

(a) Comparison among Regular, DPSGD, PSAC, and DP-Gaussian training.

(b) Comparison among DP-Gaussian, DP-SmoothAdv, DP-Stability, and DP-MACER.

Figure 5: Per-sample metric comparison, MNIST, $\sigma = 0.5$

the feature extractor to perturbations of the data. This metric was initially proposed by Yang et al. (2020) to investigate adversarial robustness and is given by

$$\frac{1}{n} \sum_{i=1}^{n} \max_{x_i' \in B_\infty(x_i, \zeta)} \frac{\|f(x_i) - f(x_i')\|_1}{\|x_i - x_i'\|_\infty}, \tag{7}$$

where the maximum is approximated in the same manner as is used for adversarial example generation, typically projected gradient descent (PGD) (Madry et al., 2018).

**RQ1:** *How are training dynamics of various methods reflected in their metric distributions?*
We calculate the three metrics for each test set data point, visualize their distribution in histograms, and compare across baselines and our proposed methods. For a detailed analysis we focus on a single setting here – MNIST and $\sigma = 0.5$ in Figure 5. The histograms for different datasets and $\sigma$'s in Figures 10-14 in Appendix D.3 follow a similar pattern and reinforce our analysis presented here. In Figure 5a, Regular training results in an approximately log-normal distribution for input gradient norms with a mode far greater than for DPSGD variants. Meanwhile, DPSGD is bimodal with some inputs having very large gradient norms which are potentially vulnerable to adversarial examples. This likely arises as a consequence of the clipping employed in the DPSGD training algorithm which effectively down-weights the contributions of hard examples and up-weights the contribution of easy examples (Shamsabadi & Papernot, 2023). Rarer samples, which would dominate a minibatch gradient for Regular training, are not learned and still have large input gradients at the end of training. PSAC mitigates this issue slightly by explicitly up-weighting hard examples, resulting in a distribution closer to Regular training. DP-Gaussian, on the other hand, shifts the distribution towards lower norm values. Comparing variants of DP-CERT in Figure 5b, DP-Stability has significantly higher input gradient norms, input Hessian spectral norms and lower local Lipschitz constants than the other three variants. This echoes the observation that TRADES-style training (Zhang et al., 2019) results in significantly lower local Lipschitz constants (Yang et al., 2020).

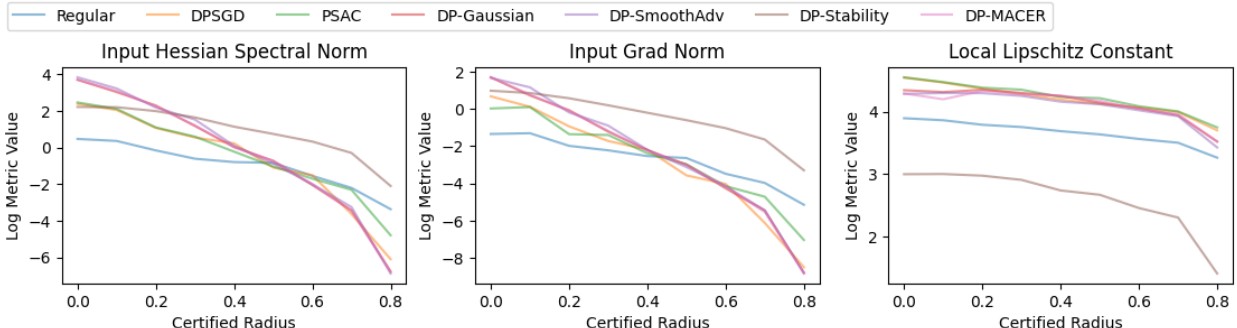

Figure 6: The input Hessian spectral norm (left), input gradient norm (middle), and local Lipschitz constants (right) calculated at different certified radii, MNIST, $\sigma = 0.5$.

**RQ2:** *How do the metrics correlate with the certified radius on a per-sample basis?* We visualize the correlation between certified radius and per-sample metrics in Figure 6. We first group examples by their certified radii, then for each group we compute their average metric values and take the logarithm. Across training methods, we see a clear negative correlation between the log metric values and certified radius, which means that examples robust to input noise tend to have lower metric values. However, different methods exhibit different levels of correlation, which is closely related to their average metric values. For example, DP-Stability on average has a much higher input gradient norm and a much lower local Lipschitz constant than other methods at the same certified radii. We further analyze the distribution of individual samples based on certified radius in Appendix D.3. We find that the local Lipschitz constant of test examples has a closer connection to CR than the gradient norm, or Hessian spectral norm for the models trained with DP guarantees. Combining this result with Figure 5a we see that DP-CERT improves certified robustness by reducing the number of datapoints with large local Lipschitz constant, input Hessian spectral norm, input gradient norm compared to PSAC and DPSGD.

## 6 Conclusion

We achieve better certified robustness with DPSGD training through augmentations, regularization, and randomized smoothing, reconciling two crucial objectives for trustworthy ML, namely privacy and robustness. We rely on DP training with augmentations that does not incur additional privacy costs, while employing various regularizations, and adversarial training methods to enhance robustness. Our resulting DP-CERT framework is modular and supports multiple combinations of these methods. Through our extensive experimental study, we confirm that DPSGD training alone, even with state-of-the-art convergence, does not provide satisfactory certified robustness. However, introducing a small number of computationally inexpensive augmentations into training by adding Gaussian noise suffices to yield strong privacy protection and certified robustness, even surpassing much more complex prior methods. By thoroughly analyzing per-sample metrics, we show that the certified radius correlates with the local Lipschitz constant and smoothness of the loss surface; this opens a new path to diagnosing when private models will fail to be robust. In conclusion, our proposed method greatly simplifies the existing solutions to simultaneously achieve CR and DP, and our practical recommendations provide a valuable contribution toward trustworthy ML. When training from scratch, Gaussian augmentations (not adversarial) should be used with DPSGD, and randomized smoothing applied at inference time. For fine-tuning pretrained models, adding stability regularization also helps accuracy, and leads to much lower local Lipschitz constants.

**Acknowledgments.** DG, FB, and NP would like to acknowledge sponsors who support their research with financial and in-kind contributions: CIFAR through the Canada CIFAR AI Chair, NSERC through a Discovery Grant, the Ontario Early Researcher Award, and the Sloan Foundation. Resources used in preparing this research were provided, in part, by the Province of Ontario, the Government of Canada through CIFAR, and companies sponsoring the Vector Institute.

**Broader Impact Statement**

Nowadays, machine learning finds extensive application in our society. As a result, ensuring the integrity of the models we build is vital. This entails safeguarding individuals' privacy while maintaining the models' robustness. Our paper focuses on examining the intersection of privacy and robustness for machine learning models, both of which are essential for establishing trustworthy ML. We introduce the DP-CERT framework, which, compared to prior work, simplifies the process for machine learning developers to create models that are both robust and private, while increasing the robustness afforded for a given privacy level.

While both differential privacy and certified robustness can provide mathematical guarantees on the privacy and robustness of trained models, the guarantees are probabilistic and can randomly fail. For DP, failures roughly occur with probability $\delta$, while for CR, some specific datapoints may not be certifiable for adequately large radii $r$. Practitioners should evaluate their risk tolerances for failures of privacy and robustness, and choose the parameters of these guarantees appropriately.

Furthermore, it has been noted that differential privacy can have particularly negative impacts on the *fairness* of models (Bagdasaryan et al., 2019; Xu et al., 2021; Esipova et al., 2023). While we do not specifically address fairness aspects in this paper, it is another prong of trustworthy ML that should be taken into account and ensured for any ML model that is to be deployed.

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

# A    Differential Privacy Background

## A.1    Differential Privacy and DPSGD

Differential Privacy (DP) (Dwork et al., 2006) is a formal framework that aims to provide mathematical guarantees on the privacy of individual data points in a dataset, while allowing one to learn properties over the entire dataset. More formally a randomized mechanism $M$ fulfills $(\varepsilon, \delta)$-DP if, for all pairs $D$ and $D'$ of neighbouring datasets (i.e. datasets differing in only one data point) and all sets $S$ of output values of $M$,

$$\Pr\left[M(D) \in S\right] \le e^{\varepsilon}\Pr\left[M(D') \in S\right] + \delta, \tag{8}$$

where $\varepsilon$ and $\delta$ are privacy parameters. $\varepsilon$ indicates the level of obtained privacy, with a smaller value of $\varepsilon$ indicating a stronger privacy guarantee. $\delta$ accounts for the probability that the algorithm will suffer a catastrophic privacy failure, which occurs when the privacy loss random variable becomes infinite (Kifer et al., 2022). This will happen when an event with zero probability for one dataset has non-zero probability for a neighbouring dataset, meaning that there is no possible multiplicative factor $e^{\varepsilon}$ that could satisfy the DP inequality Eq. 8 with $\delta = 0$. Hence, a larger value of $\delta$ increases the possibility of unmitigated privacy leakage.

The most popular algorithm for implementing DP guarantees in ML is differentially private stochastic gradient descent (DPSGD) (Abadi et al., 2016). It extends the standard SGD algorithm with two additional steps, namely gradient clipping and noise addition. While the former bounds the sensitivity of the model update, the latter implements the privacy guarantee by preventing the gradients from revealing too much information about individual data points. More formally, let $\theta_t$ denote the model parameters at training iteration $t$. At each iteration $t$, DPSGD computes the gradient of the loss function with respect to $\theta_t$ at an individual data point $x$ as $\nabla_\theta L(\theta_t, x)$. The data point's gradient is then clipped to a maximum norm $C$ using the operation clip $(\nabla_\theta L(\theta_t, x), C)$, which replaces the gradient with a vector of the same direction but smaller magnitude if its norm exceeds $C$. The clipped gradient from all datapoints in a batch are aggregated, then perturbed by adding random noise from $\mathcal{N}(0, \rho^2 C^2 \mathbf{I})$, where $\rho$ is the noise scale parameter. We detail DPSGD in Algorithm 1.

## A.2    PSAC

Per-sample adaptive clipping (PSAC) (Xia et al., 2023) is one of a number of approaches that tries to reduce the bias from per-example clipping (Bu et al., 2023a;b; Esipova et al., 2023), and was motivated for maximizing the signal to noise ratio of gradient updates. It has shown the best performance for differentially private models on several datasets including MNIST, Fashion-MNIST and CIFAR10. To complement DPSGD as an algorithm with better convergence, we incorporated PSAC into our experiments.

PSAC is similar to DPSGD, with the exception that it employs a different clipping method,

$$\text{clip}_{C,r}(g_{i,t}) = C \cdot g_{i,t} \Big/ \left( \|g_{i,t}\| + \frac{r}{\|g_{i,t}\| + r} \right). \tag{9}$$

Where $g_{i,t}$ denotes the loss gradient for the $i$th sample at iteration $t$. The motivation of this clipping method is that per-example gradients with small norms come from data points on which the model has already converged, and these gradients are often orthogonal to mini-batch gradients. Hence, small norm gradients should not have disproportionally large contributions to the batch gradient as when using per-sample normalization methods (Bu et al., 2023b). Compared to such approaches, PSAC reduces the influence of small norm gradients, and empirically shows better convergence and utility.

# B    Additional Background and Related Work

## B.1    Randomized Smoothing

Previous works have tackled improving the certified robustness of randomized smoothing methods in a variety of ways. The dominant approach for doing so involves modifications to training the base classifier so as to

increase robustness and accuracy under Gaussian perturbations. The simplest approach involves adding noise to inputs during training (Cohen et al., 2019), while other works utilize regularization (Zhai et al., 2020; Li et al., 2019; Jeong & Shin, 2020), ensembling (Horváth et al., 2022), and adversarial training (Salman et al., 2019). While these modifications have been independently studied in the context of improving the certified accuracy of randomized smoothing classifiers, our work is the first to integrate these methods with private training through augmentation multiplicity. We provide additional information on two of these training modification methods mentioned in Section 3, as well as Consistency regularization used in one ablation experiment.

### B.1.1 SmoothAdv

One of the most effective methods for improving the performance of a randomized smoothing classifier utilizes adversarial training of the classifier. The method, SmoothAdv proposed in (Salman et al., 2019), was motivated by the idea that to improve certified accuracy at a larger certification radius one needs a classifier that is more robust to local perturbations, and the best known method of achieving that is through adversarial training.

Given a soft classifier $F : \mathbb{R}^d \to P(Y)$ where $P(Y)$ is the set of probability distributions over $Y$, its smoothed soft classifier $G$ is defined as:

$$G(x) = (F * \mathcal{N}(0, \sigma^2 I))(x) = \mathbb{E}_{\eta \sim \mathcal{N}(0, \sigma^2 I)}[F(x + \eta)]. \tag{10}$$

The goal of SmoothAdv is to find a point $\hat{x}$ that maximizes the cross entropy loss of $G$ in an $l_2$ ball around $x$. SmoothAdv uses a projected gradient descent variant to approximately find $\hat{x}$, and define $J(x') = L_{\text{CE}}(G(x'), y)$ to compute

$$\nabla_{x'} J(x') = \nabla_{x'} \left( -\log \mathbb{E}_{\eta \sim \mathcal{N}(0, \sigma^2 I)}[F(x' + \eta)_y] \right). \tag{11}$$

Since the expectation in Equation 11 is difficult to compute exactly, a Monte Carlo approximation is used by sampling noise $\eta_1, ..., \eta_m \sim \mathcal{N}(0, \sigma^2 I)$ to approximately compute $\nabla_{x'} J(x')$,

$$\nabla_{x'} J(x') \approx \nabla_{x'} \left( -\log \left( \frac{1}{m} \sum_{i=1}^{m} [F(x' + \eta_i)_y] \right) \right). \tag{12}$$

Finally, $x'$ is updated by taking a step in the direction of $\nabla_{x'} J(x')$, and the final $x'$ is used to train the classifier.

### B.1.2 Consistency Regularization

For ablation purposes, we also compare to Consistency regularization (Jeong & Shin, 2020), a method very similar to Stability training which instead minimizes the KL divergence between $\hat{F}_\theta(x_i)$ and $F_\theta(x_i)$, where $\hat{F}_\theta(x_i) = \frac{1}{K} \sum_j F_\theta(x_i^j)$ is the average output probability of all smoothed samples. The loss can be expressed as

$$L_{\text{cons}}(x_i, y_i) = \sum_j L_{\text{CE}}(x_i^j, y_i) + \gamma D_{\text{KL}}\big(\hat{F}_\theta(x_i) || F_\theta(x_i^j)\big). \tag{13}$$

### B.1.3 MACER

Similar to the Stability training method, MACER (Zhai et al., 2020) also modifies the loss for optimization so that the final model has higher certified accuracy at a larger certified radius. In contrast to SmoothAdv, regularizing the model to be more robust in this sense does not require generation of adversarial examples and instead can be optimized directly. The method achieves this by decomposing the error of the smoothed classifier into a classification error term and a robustness error term. The former captures the error from the smoothed classifier misclassifying a given datapoint and the latter captures the error of a certified radius being too small.

Robustness error, much like hard label classification error, cannot be optimized directly. To address this, MACER proposes a surrogate loss to minimize the robustness error term—a hinge loss on the data $(x, y)$ for which $g_\theta(x) = y$,

$$\max\{0, \gamma - (\Phi^{-1}(\hat{f}_\theta(x)_y) - \Phi^{-1}(\hat{f}_\theta(x)_{\hat{y} \neq y}))\}, \tag{14}$$

where $\hat{f}_\theta(x)$ denotes the average of softmax probabilties on Gaussian perturbations of $x$, $\hat{f}_\theta(x)_y$ denotes the softmax probability $\hat{f}_\theta(x)$ assigned to the true class $y$, and $\hat{f}_\theta(x)_{\hat{y} \neq y}$ denotes the maximum softmax probability over classes other than the true class. This loss term is added as a regularization term to the cross entropy loss from the soft smoothed classifier.

### B.2 DPSGD and Robustness

The adversarial robustness of differentially private models has been studied in several prior works. Tursynbek et al. (2021) demonstrated that models trained with DPSGD are sometimes more vulnerable to input perturbations. Boenisch et al. (2021) further consolidate this claim with more experiments, and showed that improper choices of hyperparameters can lead to gradient masking. Zhang & Bu (2022) found that the success of adversarially training robust models with DPSGD depends greatly on choices of hyperparameters, namely smaller clipping thresholds and learning rates, differing from those that produce the most accurate models. These works reveal interesting adversarial robustness characteristics of DP models, however, they do not endeavor to *improve* the robustness of DP models.

Bu et al. (2022a) propose combining adversarial training with DPSGD. They achieve this by replacing the original example with an adversarially crafted example, obtained with a FGSM or PGD attack. Their work is orthogonal to ours in that they focus on the robustness to adversarial attacks, whereas we focus on certified robustness. Additionally, they do not perform any data augmentation, which has been shown to be effective against adversarial attacks (Rebuffi et al., 2021).

Compared with existing works in DP and CR (Phan et al., 2019; 2020; Tang et al., 2022) reviewed in Section 2, DP-CERT 1) presents a simple and effective training scheme with augmentations, allowing practitioners to introduce different adversarial training techniques through noising, regularization, and adversarially crafted examples, 2) does not rely on a denoiser at inference time, reducing inference latency, and 3) can be used for training both randomly initialized or pre-trained networks.

## C Experimental Details

### C.1 Datasets

We conduct experiments on MNIST, Fashion-MNIST and CIFAR10. The MNIST database of handwritten digits has a training set of 60,000 examples, and a test set of 10,000 examples, as does Fashion-MNIST. Each example in MNIST and Fashion-MNIST is a 28×28 grayscale image, associated with one label from 10 classes.

The CIFAR10 dataset consists of 60,000 RGB images from 10 classes, with 6,000 images per class. There are 50,000 training images and 10,000 test images of size 32×32×3.

### C.2 Code and Implementation

Our implementation of DP-CERT in code is provided as supplementary material. We give credit to the original repository for the implementation of SmoothAdv, MACER, Stability, and Consistency regularization.[1] For CERTIFY[2] and computing local Lipschitz constants[3], we use the code provided by the original authors. All experiments were conducted on a cluster of 8 Nvidia V100 GPUs. All the training and inference

---

[1]https://github.com/jh-jeong/smoothing-consistency
[2]https://github.com/locuslab/smoothing
[3]https://github.com/yangarbiter/robust-local-lipschitz

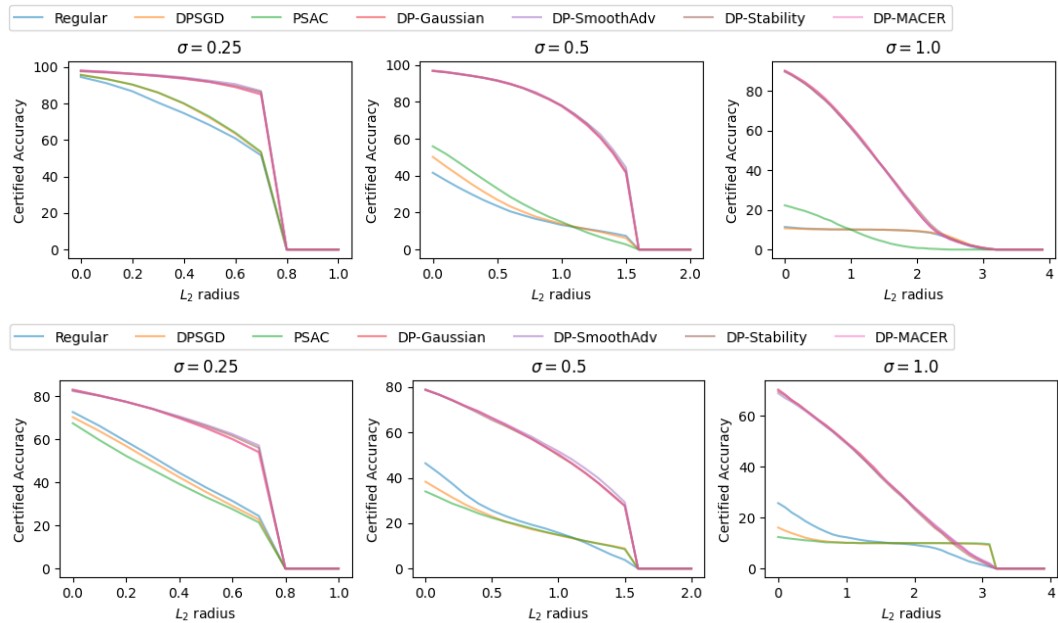

Figure 7: Approximate certified accuracy comparison on MNIST (top), and Fashion-MNIST (bottom).

procedures are implemented based on Pytorch v1.13.0 (Paszke et al., 2019) and Opacus v1.3.0 (Yousefpour et al., 2021), and our code is provided at github.com/layer6ai-labs/dp-cert.

### C.3 Additional Implementation and Hyperparameters

We set the learning rate as 0.001 and train the models for 10 epochs. The rest of the hyperparameters are the same as used by Bu et al. (2022b). For evaluation, we use CERTIFY with parameters $n = 10,000$, $n_0 = 100$, and $\alpha = 0.001$, following previous work (Cohen et al., 2019; Salman et al., 2019). Our implementations of DP-SmoothAdv, DP-Stability and DP-MACER are adapted from the original codebases for those regularization approaches (Salman et al., 2020; Li et al., 2019; Zhai et al., 2020), and we use the default hyperparameters reported in the original papers. We set the number of augmentations $K$ to 2 for MNIST and Fashion-MNIST, and 1 for CIFAR10, as they bring a better trade-off between certified accuracy and efficiency (see Figure 4).

## D Additional Results

In this section, we show the complete results and figures for our comparative study, ablations, and analysis of fine-grained metrics.

### D.1 Additional Comparative Study

Figure 7 complements Figure 2 from the main text by comparing the approximate certified accuracy between DP-CERT and baselines on MNIST, and Fashion-MNIST. The results confirm that Regular and DP training methods that do not specifically consider robustness as a criteria give poor CR results, but all instantiations of the DP-CERT framework do much better. We are unable to show further results similar to Figure 3 using the baselines SecureSGD, StoBatch, and TransDenoiser because their implementations are highly customized and not compatible with our codebase based on Pytorch and Opacus. The baseline results in Figure 3 are directly copied from (Tang et al., 2022) which provided limited experimental settings for us to match.

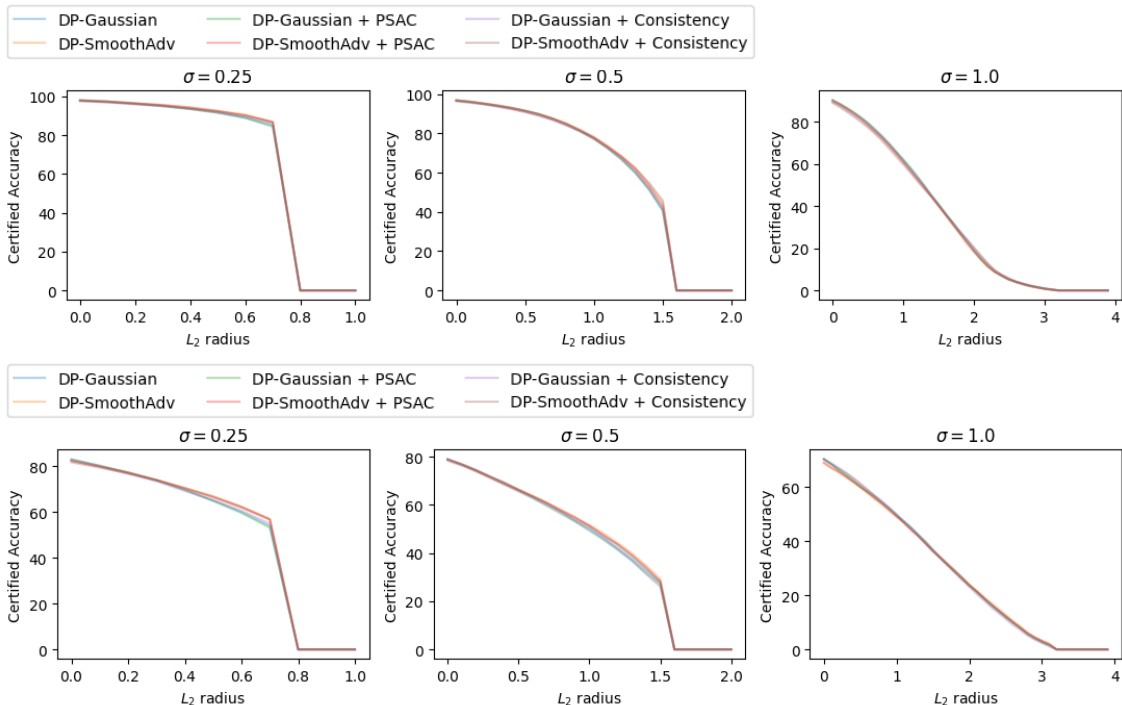

Figure 8: Certified accuracy of DP-Gaussian and DP-SmoothAdv combined with consistency regularization and PSAC on MNIST (top) and Fashion-MNIST (bottom).

## D.2 Additional Ablation Study

To extend Figure 4 from the main text, we complete the ablation study in Figures 8 and 9 on MNIST and FashionMNIST under $\sigma \in \{0.25, 0.5, 1.0\}$. Figure 8 shows the effects of adding Consistency regularization, or changing the DP clipping method to PSAC. Neither modification leads to significant changes to the certified accuracy, so we recommend using the simplest combination of standard DPSGD clipping without additional regularization when training models from scratch. Figure 9 shows the effects of changing the multiplicity of augmentations $K$. Since additional augmentations greatly increase computation time, we must navigate the tradeoff between efficiency and performance. We see that it is beneficial to use some number of augmentations or replace the original datapoint with a single augmentation (marked line 0), but beyond $K = 2$ there is little further gain in certified accuracy. We note that the case without augmentations at all, namely DPSGD, achieves much worse CR as presented in Table 2 and Figures 2 and 7. One of our main conclusions is that adding a small number of Gaussian augmentations to DPSGD is sufficient to improve certified robustness at a low cost to efficiency.

Table 3 and 4 show additional hyperparameter studies on learning rate $\lambda$ and clipping bound $C$ for DP-Gaussian under $\sigma = 0.25$. Not only does this show that our models were well-tuned for clean accuracy, but that there is no dramatic drop in performance for nearby parameter values.

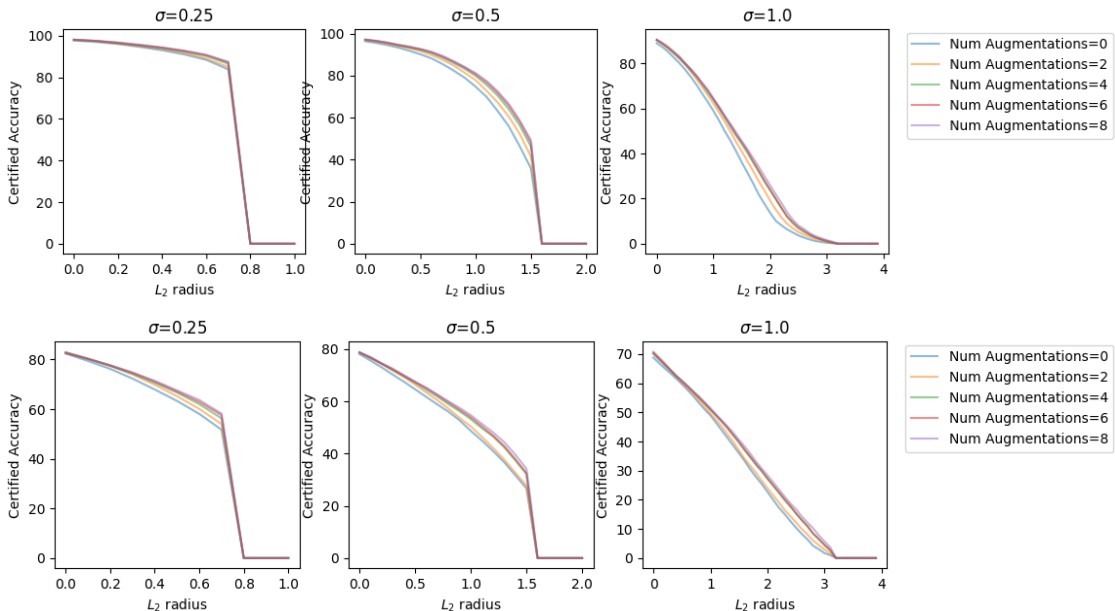

Figure 9: Certified accuracy comparison under different numbers of augmentations (0, 2, 4, 6 and 8) on MNIST (top) and Fashion-MNIST (bottom). The line marked 0 indicates replacing the original datapoint with an augmentation.

Table 3: Effect of Learning Rate $\lambda$ on Clean Accuracy (%)

| Dataset | Learning Rate $\lambda$ | Clean Accuracy (%) |
|---|---|---|
| MNIST | 0.25 | 97.71 |
| | **0.5** | **98.13** |
| | 1 | 97.78 |
| Fashion-MNIST | 2 | 83.47 |
| | **4** | **84.76** |
| | 8 | 83.96 |
| CIFAR10 | 0.0005 | 84.11 |
| | **0.001** | **87.61** |
| | 0.002 | 85.53 |

## D.3 Additional Per-sample Metric Analysis

Figures 10 through 14 complete our presentation of the three proposed metrics for interpreting robustness, the input gradient norms, input Hessian spectral norms, and local Lipschitz constants, over three datasets and values of $\sigma$ (see also Figure 5 in the main text). We use various training methods on MNIST and Fashion-MNIST under $\sigma \in \{0.25, 0.5, 1.0\}$. Echoing the analysis of RQ1 in Section 5.2, DPSGD produces a bimodal distribution for the Hessian and gradient norms, while Regular training exhibits a log-normal distribution and smaller tails for large metric values. PSAC shifts the distributions to be closer to those of Regular training by reducing clipping bias. DP-CERT methods, on the other hand, shift the distribution towards smaller metric values, resulting in higher certified accuracy. An exception is DP-Stability, which has significantly higher average gradient and Hessian norms, but without the mode at very high values, and with lower local Lipschitz constants than the other three variants.

Table 4: Effect of Clipping Norm $C$ on Clean Accuracy (%)

| Clipping Norm $C$ | Clean Accuracy (%) | | |
|:---:|:---:|:---:|:---:|
| | MNIST | Fashion-MNIST | CIFAR10 |
| 0.01 | 94.89 | 75.60 | 83.79 |
| 0.05 | 97.75 | 83.42 | 83.99 |
| **0.1** | **98.13** | **84.76** | **87.61** |
| 0.2 | 97.72 | 84.00 | 84.19 |
| 0.5 | 96.47 | 81.46 | 84.24 |

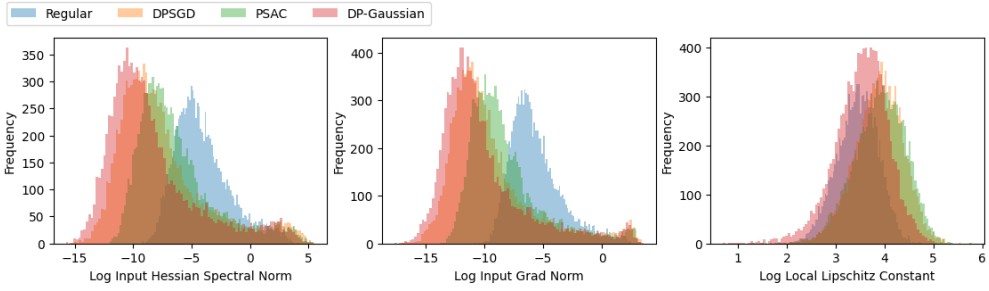

(a) Comparison of Regular, DPSGD, PSAC, and DP-Gaussian training.

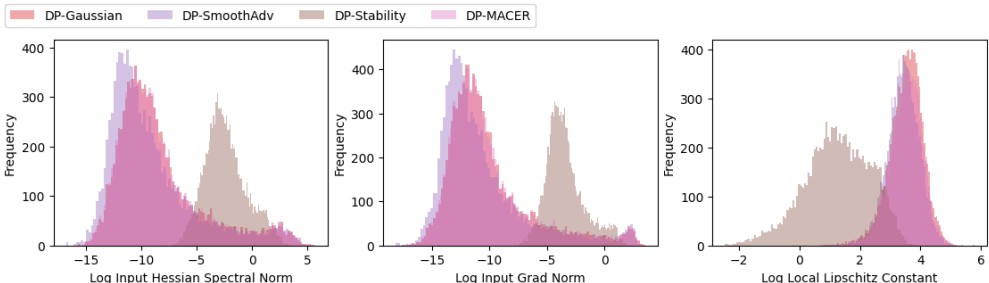

(b) Comparison of DP-Gaussian, DP-SmoothAdv, DP-Stability, and DP-MACER.

Figure 10: Per-sample metric comparison, MNIST, $\sigma = 0.25$

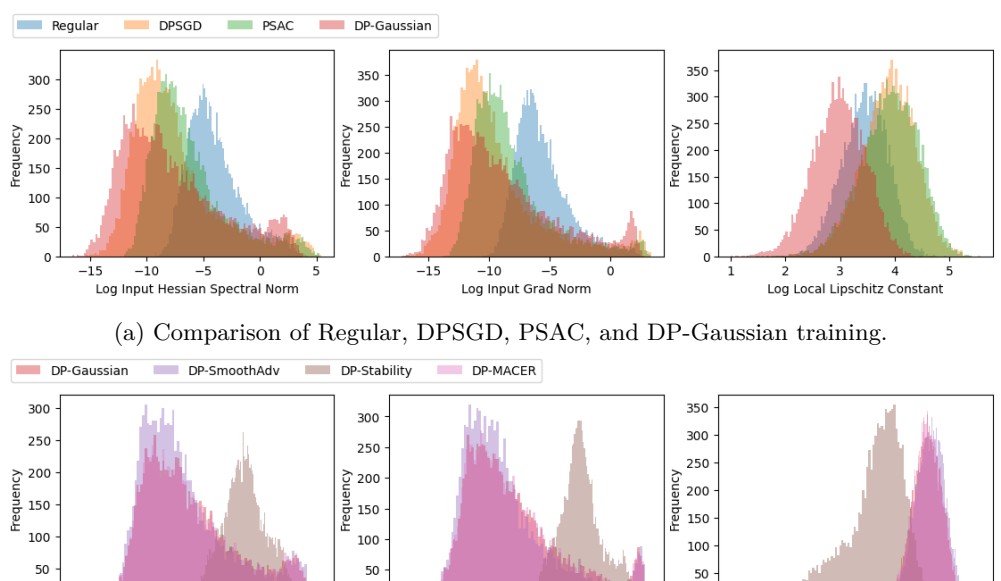

(a) Comparison of Regular, DPSGD, PSAC, and DP-Gaussian training.

(b) Comparison of DP-Gaussian, DP-SmoothAdv, DP-Stability, and DP-MACER.

Figure 11: Per-sample metric comparison, MNIST, $\sigma = 1.0$

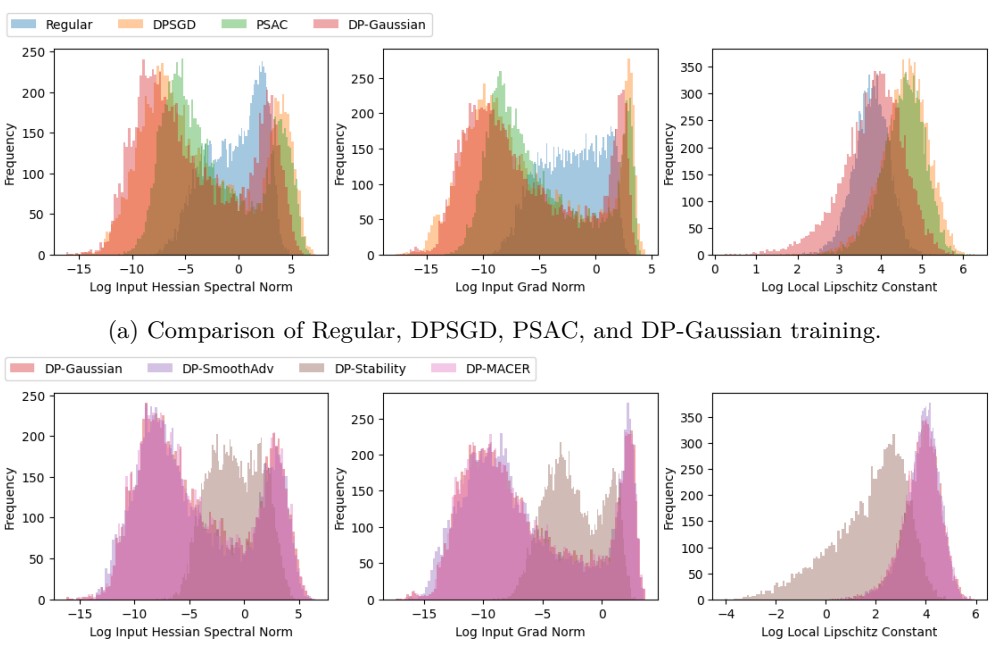

(a) Comparison of Regular, DPSGD, PSAC, and DP-Gaussian training.

(b) Comparison of DP-Gaussian, DP-SmoothAdv, DP-Stability, and DP-MACER.

Figure 12: Per-sample metric comparison, Fashion-MNIST, $\sigma = 0.25$

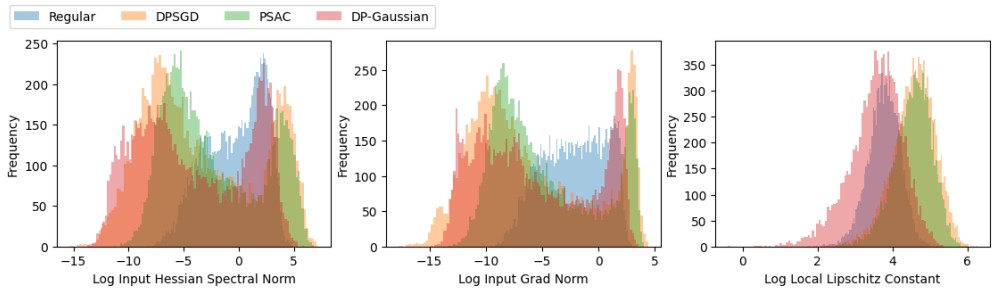

(a) Comparison of Regular, DPSGD, PSAC, and DP-Gaussian training.

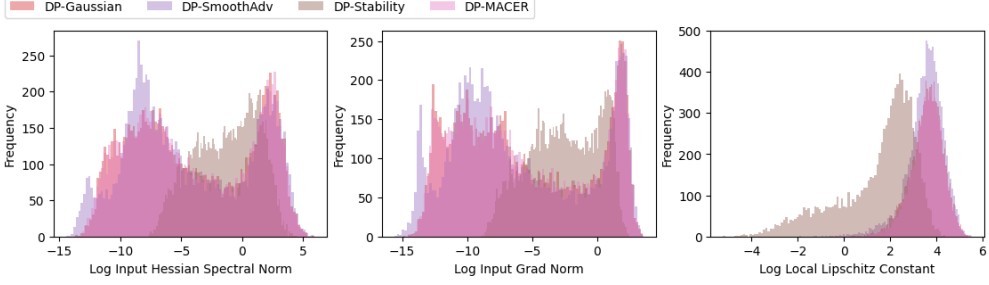

(b) Comparison of DP-Gaussian, DP-SmoothAdv, DP-Stability, and DP-MACER.

Figure 13: Per-sample metric comparison, Fashion-MNIST, $\sigma = 0.5$

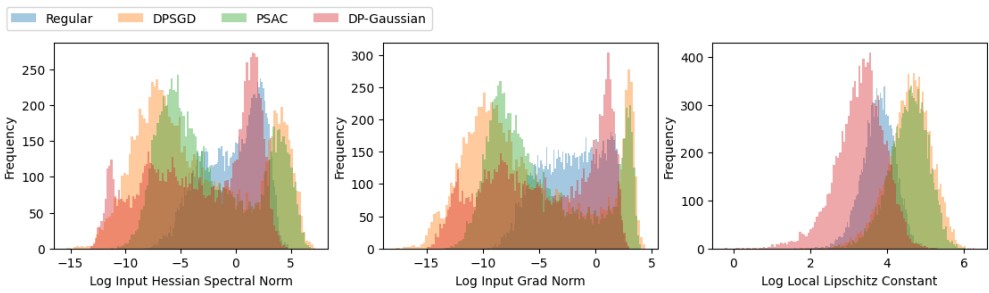

(a) Comparison of Regular, DPSGD, PSAC, and DP-Gaussian training.

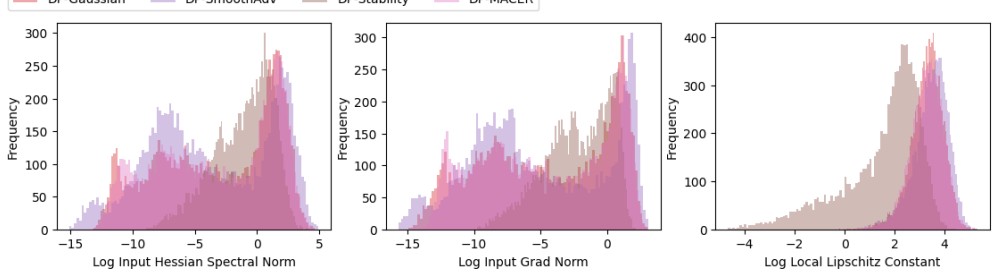

(b) Comparison of DP-Gaussian, DP-SmoothAdv, DP-Stability, and DP-MACER.

Figure 14: Per-sample metric comparison, Fashion-MNIST, $\sigma = 1.0$

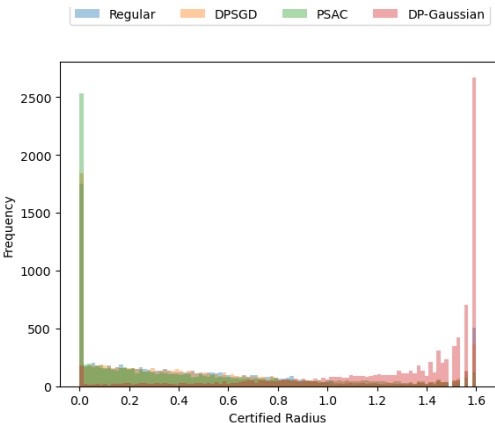

Figure 15: Distributions of certified radii for Regular, DPSGD, PSAC, and DP-Gaussian training on MNIST with $\sigma = 0.5$.

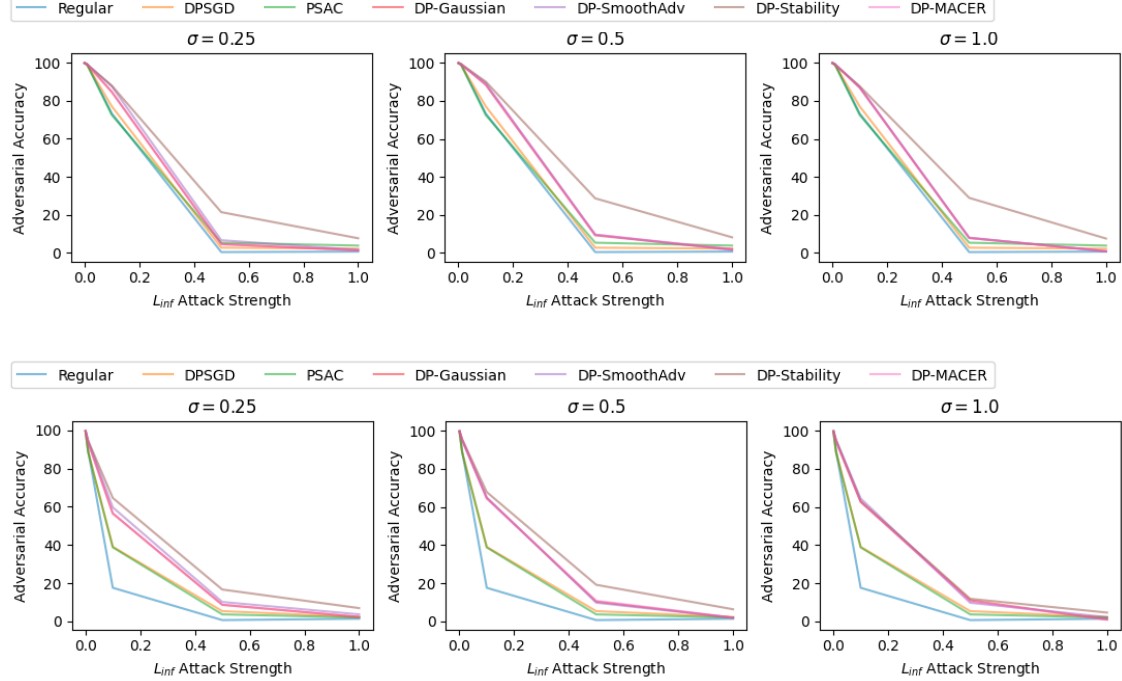

Figure 16: Adversarial accuracy against a $L_\infty$-FGSM attack under various attack strengths on MNIST (top) and Fashion-MNIST (bottom).

Figure 15 shows the distribution of certified radii for baseline training methods and an instance of DP-CERT for the same settings as Figure 5. Whereas Regular, DPSGD, and PSAC training all have a large spike of samples that cannot be certified at any level, DP-Gaussian achieves certified radii above 1.0 for most samples, with the mode even higher at 1.6.

While our main focus has been on certified robustness, we briefly examine adversarial robustness. Figure 16 shows the adversarial accuracy under a $l_\infty$-FGSM attack, with the attack strength in $\{0.0005, 0.01, 0.1, 0.5, 1\}$. Consistent with the ranking of the average local Lipschitz constant from Figures 10 through 14, DP-Stability consistently outperforms other approaches, while DP-Gaussian, DP-SmoothAdv, and DP-MACER all achieve similar adversarial accuracy above that of the unprotected baselines.

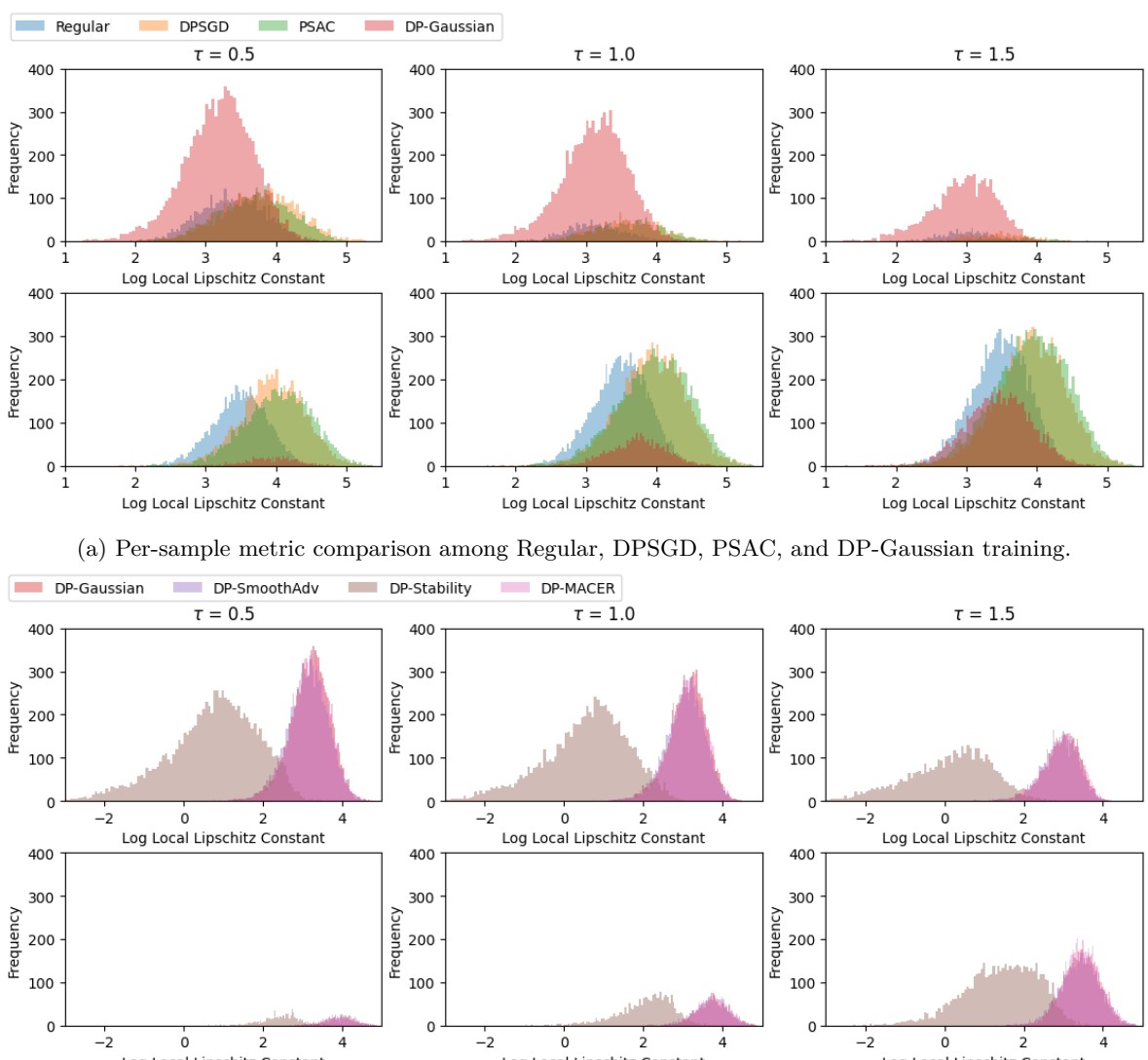

(a) Per-sample metric comparison among Regular, DPSGD, PSAC, and DP-Gaussian training.

(b) Per-sample metric comparison among DP-Gaussian, DP-SmoothAdv, DP-Stability, and DP-MACER.

Figure 17: Distributions of log local Lipschitz constants among baselines and proposed methods, MNIST, $\sigma = 0.5$. In each subfigure, data points are classified into ones with certified radius above the threshold $\tau$ (top row), and below the threshold (bottom row). We display $\tau \in \{0.5, 1.0, 1.5\}$.

Finally, as shown in Figure 17, we select three certified radius thresholds $\tau \in \{0.5, 1.0, 1.5\}$, and separately plot the log of local Lipschitz constants of examples above and below $\tau$ on the top and bottom rows of the subfigures respectively. First, the examples with certified radii *below* the threshold have *higher* average local Lipschitz constant. Second, as we increase the threshold $\tau$, more examples with higher local Lipschitz constant end up below the certified radius threshold. We do a similar analysis for the other two metrics in Figures 18 and 19, but since the local Lipschitz constant is derived using a PGD attack, it naturally correlates with the robustness to adversarial examples better. For further comparison, we present the FGSM accuracy on MNIST and Fashion-MNIST under attack strengths in $\{0.0005, 0.01, 0.1, 0.5, 1\}$ in Figure 16. Consistent with the ranking of the average local Lipschitz constant, DP-Stability consistently outperforms other approaches, while DP-Gaussian, DP-SmoothAdv and DP-MACER achieve similar adversarial accuracy over different attack margins.

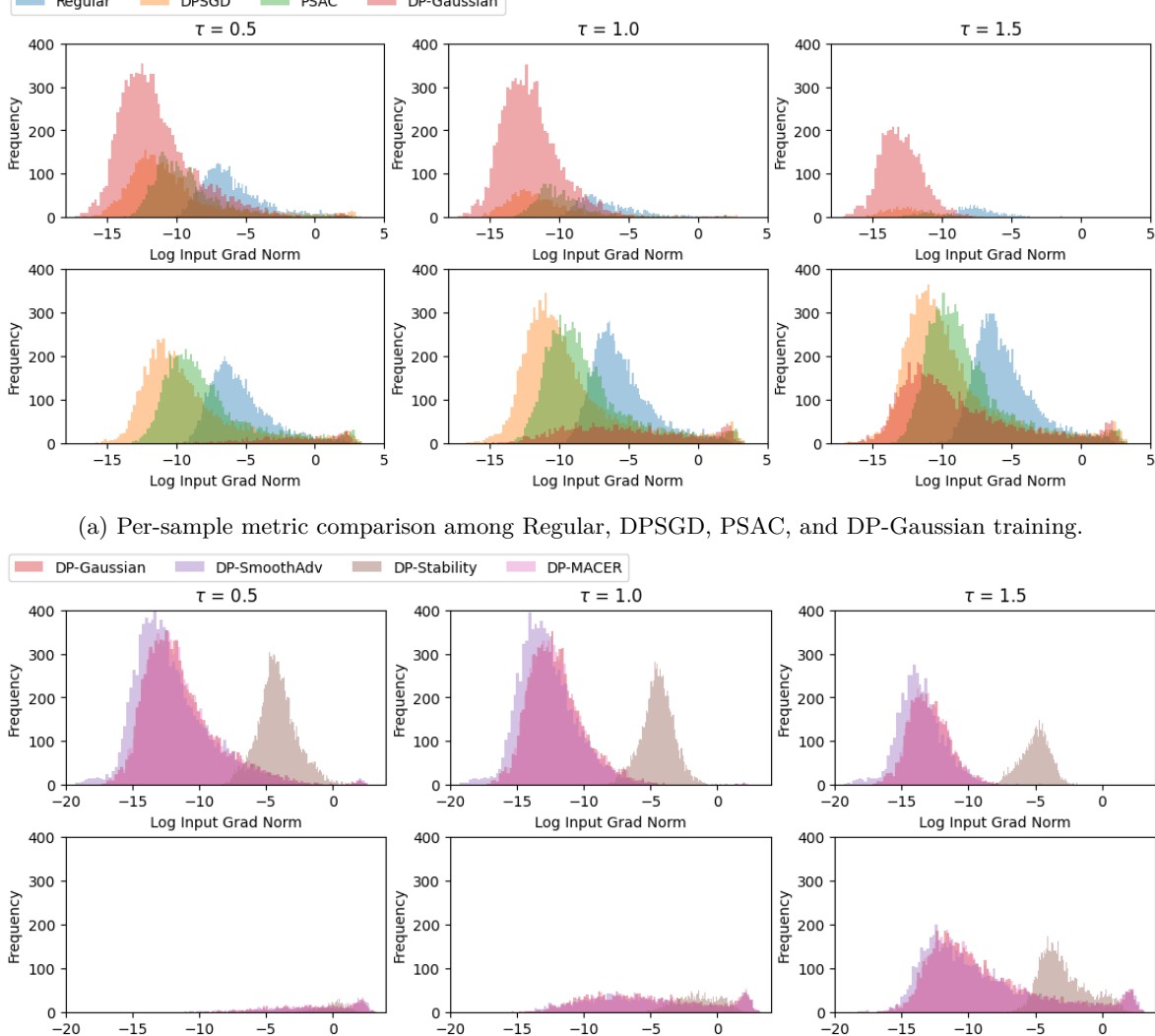

(a) Per-sample metric comparison among Regular, DPSGD, PSAC, and DP-Gaussian training.

(b) Per-sample metric comparison among DP-Gaussian, DP-SmoothAdv, DP-Stability, and DP-MACER.

Figure 18: Comparing the distribution of *input gradient norm* of baselines and proposed methods on MNIST. All DP-CERT methods are trained with $\sigma = 0.5$. We show the logarithmic metric values for all methods for better visualization.

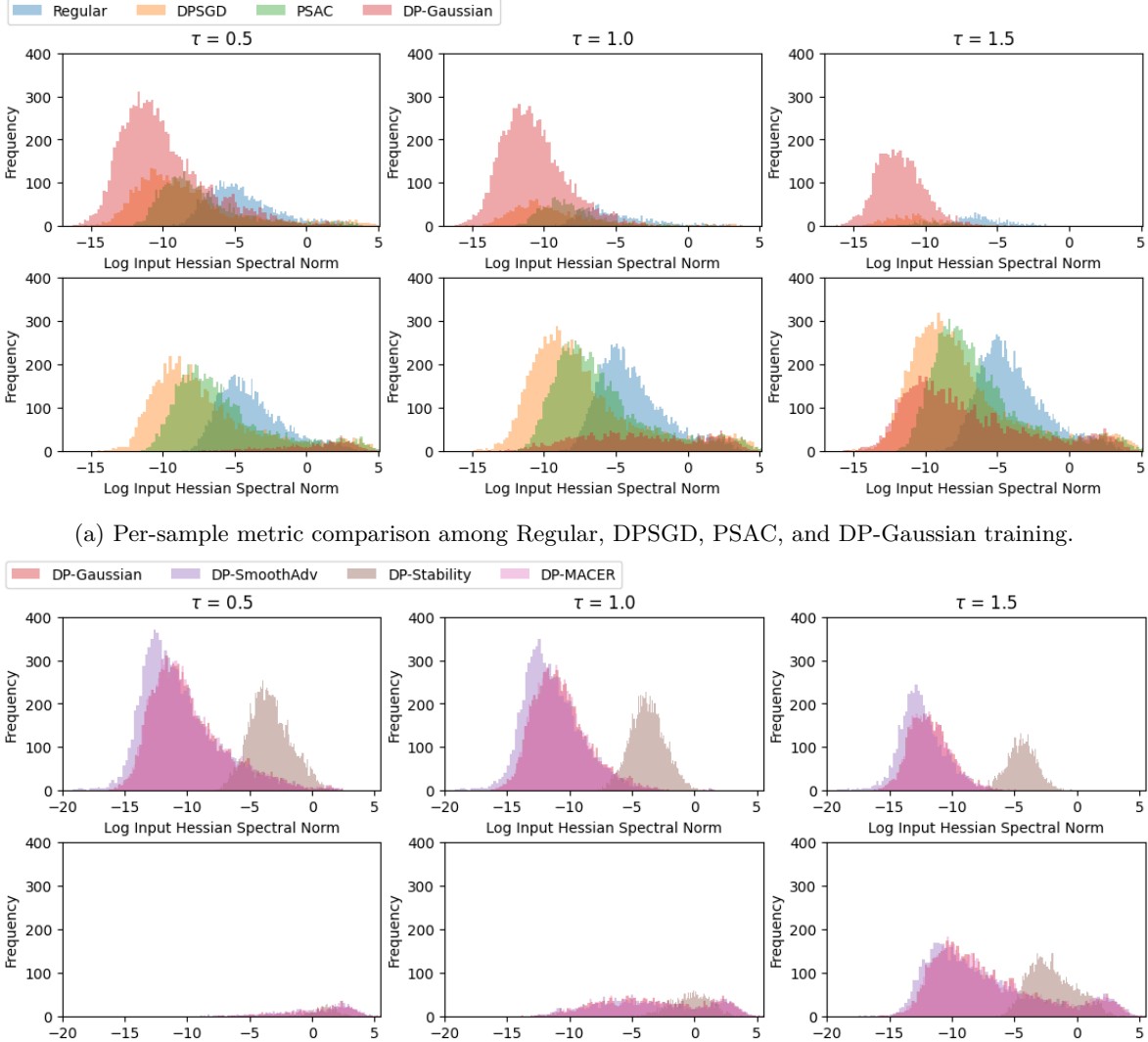

(a) Per-sample metric comparison among Regular, DPSGD, PSAC, and DP-Gaussian training.

(b) Per-sample metric comparison among DP-Gaussian, DP-SmoothAdv, DP-Stability, and DP-MACER.

Figure 19: Comparing the distribution of *input Hessian spectral norm* of baselines and proposed methods on MNIST. All DP-CERT methods are trained with $\sigma = 0.5$. We show the logarithmic metric values for all methods for better visualization.

