# OpenReview forum: "Augment then Smooth: Reconciling Differential Privacy with Certified Robustness"
_TMLR — Accepted by TMLR_

### Review · Reviewer_7N2U · 2024-04-01

**Summary Of Contributions:**

The authors presents DP-CERT, a method integrating randomized smoothing into differentially private model training. This approach significantly enhances both privacy and robustness, with a notable increase in certified accuracy on CIFAR10, compared to prior work. The paper emphasizes the importance of privacy and robustness in machine learning, challenges in achieving both, and how DP-CERT offers a simple, adaptable framework for this purpose. Through experiments, DP-CERT is shown to improve certifiable robustness without compromising privacy, offering practical guidance for trustworthy machine learning implementations.

**Audience:**

Yes

**Claims And Evidence:**

Yes

**Requested Changes:**

Expand Experimental Evaluation: Extend the evaluation to more datasets and model architectures, ensuring the framework's applicability and robustness across various domains.

Optimize Computational Efficiency: Investigate methods to reduce the computational overhead introduced by DP-CERT, making it more accessible for real-world applications.

Fairness Assessment: Conduct an analysis of the impact on model fairness when applying DP-CERT, addressing potential disparate impacts.

Hyperparameter Sensitivity Analysis: Provide a detailed sensitivity analysis of the key hyperparameters, aiding practitioners in model tuning.

**Strengths And Weaknesses:**

Innovative Approach: DP-CERT's integration of randomized smoothing into DP training is a significant advancement, addressing the long-standing conflict between achieving DP and CR in a unified model.
Impressive Performance: The method sets new benchmarks in certified accuracy on CIFAR10, substantially outperforming prior works while maintaining stringent privacy guarantees.
Comprehensive Analysis: The per-sample metric analysis offers deep insights into the relationship between certifiable radii and model robustness, guiding future efforts in enhancing CR in DP models.

Weakness:

Model Complexity: Despite its effectiveness, the DP-CERT framework introduces additional complexity in training, potentially increasing computational resources and training time.
Generalizability: The experiments primarily focus on image classification tasks and CIFAR10 dataset. Further studies are required to assess the framework's performance across diverse datasets and model architectures.
Impact on Fairness: The paper does not explore the impact of combining DP and CR on model fairness, an aspect critical for deploying AI in sensitive applications.

---

### Review · Reviewer_jh7m · 2024-05-06

**Summary Of Contributions:**

The paper proposes a method called DP-CERT that achieves better-certified robustness with DP-SGD training through augmentations, regularization, and randomized smoothing, reconciling privacy and robustness for trustworthy ML. While DP-CERT is a combination of known methods, the experiments show that it reduces the computational complexity and increases the certified accuracy compared to previous works that bring CR guarantees to DP.

**Audience:**

Yes

**Claims And Evidence:**

Yes

**Requested Changes:**

1. Since the paper focuses on studying differential privacy with certified robustness, I would suggest the authors move the introduction of  DP from the appendix to the main paper, e.g., Section 2. The present version is a little bit unclear if the audience is not familiar with DP.

2. As mentioned in the weakness, there is a lack of rigorous proof of the main results. A formal privacy guarantee for the proposed algorithm is required.  Specifically, if the method is run with a noise scale $\rho$, then it satisfies $(\epsilon,\delta)$-DP with $\epsilon=?$. It is unclear to me.

3. Experiments compare the proposed method with the state of the art. However, all methods need to be compared under the same level of privacy guarantee, e.g., the same $\epsilon$ and $\delta$. Since the privacy guarantee of the proposed method is missing, I’m not sure the results of the experiments are fair and reasonable.

4. In Table 2, when $\sigma$ increases, DP-stability and DP-MACER outperform the other methods, while when $\sigma$ is small, DP-smoothadv seems better. Are there any potential reasons or explanations?

**Strengths And Weaknesses:**

Strength:

A simple and effective method, DP-CERT, that achieves both privacy and robustness guarantees simultaneously is proposed. It has been shown in the experiments that DP-CERT outperforms the existing works in the real-world datasets.

Weakness:

1. The paper mainly proves the effectiveness of the proposed algorithm through experiments, but lacks theoretical support. Although it is an experimental study, I think that at least the theoretical results of the privacy guarantee part should be formally given as a theorem. Specifically, what level of differential privacy does the proposed algorithm satisfy when updating parameter via (2)?

2. Due to the lack of theoretical results in the privacy guarantee part, I'm not sure whether the comparison of the existing algorithms in the experiments is fair. Since even if the same noise is added in each iteration, the privacy guaranteed by each algorithm may still be different. We should compare the accuracy of different methods to achieve the same level of differential privacy (the same $\epsilon$ and $\delta$).

---

> ### Author Response · Authors · 2024-05-22
> **Response to Reviewer jh7m (1/2)**
>
> We thank the reviewer for their feedback, and are glad that they found our proposed method to be “simple and effective”, with strong performance. There were several valid questions around the privacy guarantees of DP-CERT, and fairness of our experimental setup, which we will gladly try to clarify.
>
> **RC1:** This is a helpful suggestion that will make our paper more accessible to a broader audience, thank you. We have updated our paper on OpenReview to move the discussion of DP and the DPSGD algorithm into the main text as part of the introduction.
>
> **RC2:** This seems to be your primary concern with our work, and was similar to a comment by Reviewer Wp45, which shows we may not have been clear enough in our discussion of the privacy guarantees of DP-CERT at the end of Section 3. Since this is an important point, allow us to expand on it here.
>
> DP-CERT was designed to closely follow the DPSGD algorithm. This was a deliberate choice, as we found previous methods to promote DP and CR simultaneously (Phan et al., 2019, 2020, Tang et al., 2022) were overly complicated and hard to implement, which contributed to greater difficulty in the privacy analysis. In contrast, DP-CERT attains the exact same DP guarantees as DPSGD which can be quantified using any privacy accounting package (e.g. Opacus, Tensorflow-Privacy) without modification.
> As you know, the crucial step in DPSGD that enables DP guarantees is per-sample gradient computation with clipping. This limits the effect that any single datapoint can have on the gradient update applied to the model weights, which is the entire premise of DP. Hence, to achieve the same guarantees as DP-SGD we must ensure that in DP-CERT all contributions from the training datapoint $x_i$ are encapsulated in a “per-sample” gradient which is clipped before being aggregated in a batch. We have improved the discussion in Section 3 and added an algorithm block to clarify where DP-CERT training deviates from DPSGD - namely DP-CERT (1) creates adversarial or Gaussian perturbations for each datapoint in the batch, (2) adds regularization to the loss function (to improve robustness), and (3) augmentation multiplicity ([1] De et al., 2022) is used to compute per-sample gradients over all augmented data $x_i^j$ associated to an original point $x_i$. Point (3) is exactly what is required to achieve DP when using augmented datapoints. De et al., 2022 [1] do provide a more formal discussion of the privacy guarantee from augmentation multiplicity, and since it is not our novel contribution we did not want to mislead the reader by repeating their proof.
>
> Meanwhile, for point (2), any loss function can be used with DPSGD as long as it acts pointwise to allow computing per-sample gradients. The regularization we add only involves a training datapoint and its augmentations, not other training datapoints.
> We add that after the model is trained, DP-CERT uses randomized smoothing for inference. This does not affect the privacy guarantees of the model in any way thanks to the post-processing guarantee of DP (e.g. Dwork & Roth 2014). Any operation that does not involve the training data can be applied to the model weights (to which the DP guarantees apply) without affecting the guarantee. Randomized smoothing certainly fits that description, so no modifications are needed.
> In summary, since DP-CERT achieves the exact same privacy guarantees as DPSGD thanks to the prior work on augmentation multiplicity, we did not feel that repeating this formal proof would add to our work, and we wanted to avoid claiming that it was our novel contribution. We hope that our non-technical arguments are sufficient to convince you that prior proofs apply in a straightforward manner.
>
> **RC3:** We completely agree that all methods need to use the same level of privacy guarantee to have a meaningful comparison of model performance across methods. We did provide this information in Appendix C.3, but admit that it should have been featured more prominently in the experiment section of the main text. To summarize, all methods (except Regular of course) achieve the same privacy guarantee of $\varepsilon=3.0$, $\delta=10^{-5}$, the same values used in the prior works we directly compare to (Phan et al., 2019, 2020, Tang et al., 2022). We have moved this information into Section 4 so that the reader will be informed about the level playing field used across methods. Thank you for the helpful reminder.

---

> ### Author Response · Authors · 2024-05-22
> **Response to Reviewer jh7m (2/2)**
>
> **RC4:** This is an interesting question, and we do not have an entirely conclusive answer based on the tests we have done. The most plausible explanation to us is that this is an interaction with properties of the dataset, as it is only on CIFAR-10 that DP-SmoothAdv shows a significant level of outperformance over other approaches at low $\sigma$. CIFAR-10 images are higher resolution than MNIST or Fashion-MNIST, which affects the properties of the adversarial and Gaussian perturbations crafted in high dimensional spaces $\mathbb{R^p}$. Meanwhile, DP-SmoothAdv is the only method we tested that uses adversarial examples at training time. We are guessing that since adversarial attacks are more successful compared to Gaussian noise attacks, especially for higher resolution images, for smaller values of $\sigma$, DP-SmoothAdv enhances the certified accuracy most effectively. However, for bigger values of $\sigma$ training with the same type of perturbation used for certification - namely, Gaussian noise - improves certified accuracy better.
>
> [1] "Unlocking high-accuracy differentially private image classification through scale." De, Soham, et al., 2022

---

### Review · Reviewer_Wp4S · 2024-05-08

**Summary Of Contributions:**

This paper shows a simple solution which combines DP-SGD with random smoothing which can provide both differential privacy and certified robustness.

**Audience:**

Yes

**Broader Impact Concerns:**

Both DP & CR are important aspects of learning. No concern on the broader impacts.

**Claims And Evidence:**

No

**Requested Changes:**

1. Formal proof of the differential privacy and certified robustness need to provided.
   When you do the proof, you need consider how the random smoothing can affect DP. Also, you need consider how DP may affect CR proof.

2. Please evaluate your solution with existing methods which establish DP and CR
3. The complete algorithm DPSGD and DP-CERT should be provided in the main texts.

**Strengths And Weaknesses:**

Strengths:
+ The paper is easy to follow
+ The approach is simple which can be easily implemented

Weaknesses:
+ There are no formal proof of the differential privacy and certified robustness.
   When you do the proof, you need consider how the random smoothing can affect DP. Also, you need consider how DP may affect CR proof.
   And when you consider the formal proof, I believe you will see the proposed may not work, or at least some detailed inference needs to be done to guarantee both.
+ There is no comparison with the existing methods which establish DP and CR
+ The complete algorithm DPSGD and DP-CERT should be provided

---

> ### Author Response · Authors · 2024-05-22
> **Response to Reviewer Wp45 (1/2)**
>
> We thank the reviewer for their time spent reviewing our paper, and were happy to see that they found our approach “simple” and easy to implement. This was one of our main objectives given the complicated solutions that had previously been developed. We will address each of the three points in turn.
>
> **RC1:** This concern was similar to a comment by Reviewer jh7m, which shows we may not have been clear enough in our discussion of the privacy guarantees of DP-CERT at the end of Section 3. Since this is an important point, allow us to expand on it here.
>
> Given that we are trying to combine the formal guarantees of DP and CR into a single method, it is understandable that you wanted to see more discussion of how they may interact. One of the benefits of our DP-CERT formulation is that the two formal analyses can be done entirely separately and do not interact, making the results straightforward. This is primarily due to the post-processing property of DP (e.g. Dwork & Roth 2014). In both DPSGD and DP-CERT, the DP guarantee applies to the model weights. Hence, any operation that does not involve the training data can be applied to the model weights without affecting the DP guarantee. After training, DP-CERT uses randomized smoothing to certify robustness, which does not involve training data, and so can be considered post-processing.
>
> On the other hand, the guarantees that randomized smoothing and CERTIFY provide do not make any assumptions on the properties of the underlying model, in particular, whether it carries DP guarantees or not. As seen in Equation 1 and the following discussion, randomized smoothing can be applied to any classifier $f$ independent whether it was trained with SGD, DPSGD, or our DP-CERT, and will return a certification radius $r$ that depends only on the smoothing noise level $\sigma$ and class probabilities from the model. We do note that the \emph{accuracy} of the smoothed predictions can vary greatly depending on how a model was trained - the certification only tells us that there are no adversarial inputs in a radius $r$ that change the output class, however, that output class can still be incorrect on a given input. The point of our DP-CERT method, as shown by our experiments, is that it improves model performance (approximate certified accuracy) and provides certification with larger radii (average certified radius).
>
> DP-CERT was designed to closely follow the DPSGD algorithm. This was a deliberate choice, as we found previous methods to promote DP and CR simultaneously (Phan et al., 2019, 2020, Tang et al., 2022) were overly complicated and hard to implement, which contributed to greater difficulty in the privacy analysis. In contrast, DP-CERT attains the exact same DP guarantees as DPSGD which can be quantified using any privacy accounting package (e.g. Opacus, Tensorflow-Privacy) without modification.
> As you know, the crucial step in DPSGD that enables DP guarantees is per-sample gradient computation with clipping. This limits the effect that any single datapoint can have on the gradient update applied to the model weights, which is the entire premise of DP. Hence, to achieve the same guarantees as DP-SGD we must ensure that in DP-CERT all contributions from the training datapoint $x_i$ are encapsulated in a “per-sample” gradient which is clipped before being aggregated in a batch. We have improved the discussion in Section 3 and added an algorithm block to clarify where DP-CERT training deviates from DPSGD - namely DP-CERT (1) creates adversarial or Gaussian perturbations for each datapoint in the batch, (2) adds regularization to the loss function (to improve robustness), and (3) augmentation multiplicity ([1] De et al., 2022) is used to compute per-sample gradients over all augmented data $x_i^j$ associated to an original point $x_i$. Point (3) is exactly what is required to achieve DP when using augmented data points. De et al., 2022 [1] do provide a more formal discussion of the privacy guarantee from augmentation multiplicity, and since it is not our novel contribution we did not want to mislead the reader by repeating their proof.
>
> Meanwhile, for point (2), any loss function can be used with DPSGD as long as it acts pointwise to allow computing per-sample gradients. The regularization we add only involves a training datapoint and its augmentations, not other training datapoints.
> In summary, since DP-CERT achieves the exact same privacy guarantees as DPSGD thanks to the prior work on augmentation multiplicity, we did not feel that repeating this formal proof would add to our work, and we wanted to avoid claiming that it was our novel contribution. We hope that our non-technical arguments are sufficient to convince you that prior proofs apply in a straightforward manner.

---

> ### Author Response · Authors · 2024-05-22
> **Response to Reviewer Wp45 (2/2)**
>
> **RC2:** We are somewhat confused by this requested change, and that you stated “There is no comparison with the existing methods which establish DP and CR”, since our main experiments in Section 5 directly compare DP-CERT to existing methods that establish DP, or both DP and CR. First, in Table 2 and Figure 2 we compared to DPSGD and PSAC which give identical DP guarantees as our method but do not aim to improve robustness (we applied randomized smoothing at inference time to these methods to generate CR guarantees). Then, in Figure 3 we compared to three recent methods that aim to give strong DP and CR guarantees (StoBatch, Phan et al. 2019; SecureSGD, Phan et al. 2020; TransDenoiser, Tang et al. 2022).
>
> **RC3:** This is a useful suggestion that was echoed by other reviewers. We will gladly oblige and have moved the DPSGD algorithm from the appendix up to Section 2 with an expanded discussion of DP. We also added an algorithm box to summarize DP-CERT at the end of Section 3 along with a more detailed comparison to DPSGD.
>
> [1] "Unlocking high-accuracy differentially private image classification through scale." De, Soham, et al., 2022

---

### Decision · Action_Editor_97sC · 2024-07-16

**Recommendation:** Accept as is

**Comment:**

The paper proposes an approach called DP-CERT, a simple and effective method that achieves both privacy and robustness guarantees. The main idea is to use augmentations that do not incur additional privacy costs, while employing regularizations and adversarial training methods to enhance robustness. All reviewers appreciated the straightforward and effective approach, supported by comprehensive experimental results. Although two reviewers suggested for theoretical proofs for explaining the empirical success of the proposed approach, the authors partially addressed this comment to justify the privacy guarantees by referencing De et al. (2022), as discussed on page 6 of the current version.

Overall, combining all reports from the reviewers and my own reading of the paper, it is evident that the paper provides contributions and observations that are novel in the literature, and the proposed method is sufficiently supported by experiments. Therefore, I conclude that it meets the standards of TMLR.

**Audience:**

The paper studies the topics related to difference privacy and certified robustness which will certainly attract interests from the machine learning community.

**Claims And Evidence:**

The proposed approaches are clearly supported by empirical evidence with the codes provided online.